# Enhancing Adversarial Contrastive Learning via Adversarial Invariant Regularization

**Xilie Xu**[1]*, **Jingfeng Zhang**[2,3]*†, **Feng Liu**[4], **Masashi Sugiyama**[2,5], **Mohan Kankanhalli**[1]

[1] School of Computing, National University of Singapore
[2] RIKEN Center for Advanced Intelligence Project (AIP)
[3] School of Computer Science, The University of Auckland
[4] School of Computing and Information Systems, The University of Melbourne
[5] Graduate School of Frontier Sciences, The University of Tokyo

xuxilie@comp.nus.edu.sg    jingfeng.zhang@auckland.ac.nz
fengliu.ml@gmail.com    sugi@k.u-tokyo.ac.jp
mohan@comp.nus.edu.sg

## Abstract

Adversarial contrastive learning (ACL) is a technique that enhances standard contrastive learning (SCL) by incorporating adversarial data to learn a robust representation that can withstand adversarial attacks and common corruptions without requiring costly annotations. To improve transferability, the existing work introduced the standard invariant regularization (SIR) to impose *style-independence property* to SCL, which can exempt the impact of nuisance *style factors* in the standard representation. However, it is unclear how the style-independence property benefits ACL-learned robust representations. In this paper, we leverage the technique of *causal reasoning* to interpret the ACL and propose adversarial invariant regularization (AIR) to enforce independence from style factors. We regulate the ACL using both SIR and AIR to output the robust representation. Theoretically, we show that AIR implicitly encourages the representational distance between different views of natural data and their adversarial variants to be independent of style factors. Empirically, our experimental results show that invariant regularization significantly improves the performance of state-of-the-art ACL methods in terms of both standard generalization and robustness on downstream tasks. To the best of our knowledge, we are the first to apply causal reasoning to interpret ACL and develop AIR for enhancing ACL-learned robust representations. Our source code is at https://github.com/GodXuxilie/Enhancing_ACL_via_AIR.

## 1 Introduction

The attention towards pre-trained models that can be easily finetuned for various downstream applications has significantly increased recently [17, 18, 19, 41]. Notably, foundation models [4] via self-supervision on large-scale unlabeled data, such as GPT [5] and CLAP [20], can be adapted to a wide range of downstream tasks. Due to the high cost of annotating large-scale data, unsupervised learning techniques [21, 34, 49, 16] are commonly used to obtain generalizable representations, in which standard contrastive learning (SCL) has been shown as the most effective way [9, 38, 10].

Adversarial contrastive learning (ACL) [30, 29], that incorporates adversarial data [37] with SCL [9], can yield robust representations that are both generalizable and robust against adversarial attacks [23]

---

*The first two authors have made equal contributions.
†Corresponding author.

37th Conference on Neural Information Processing Systems (NeurIPS 2023).

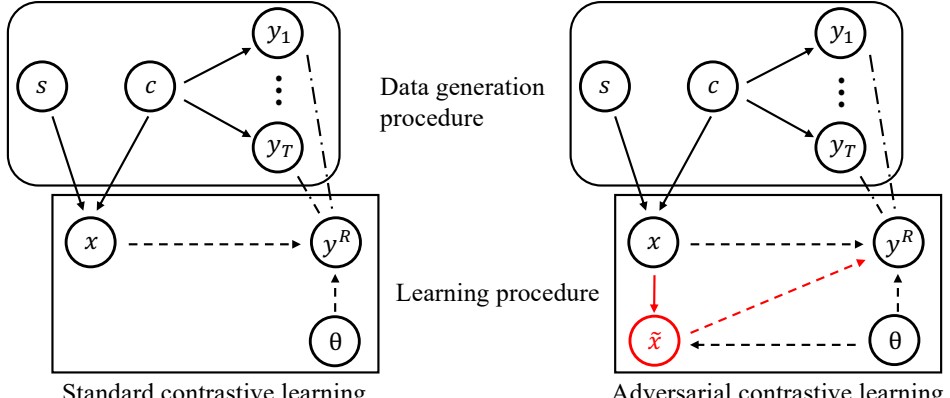

Figure 1: Causal graph of data generation procedure in SCL [38] (left panel) and ACL (right panel). $s$ is style variable, $c$ is content variable, $x$ is unlabeled data, $\tilde{x}$ is the generated adversarial data, and $\theta$ is the parameter of representation. The solid arrows are causal, but the dash arrows are not causal. The proxy label $y^R \in \mathcal{Y}^R$ is a refinement of the target labels $y_t \in \mathcal{Y} = \{y_i\}_{i=1}^T$ that represents an unknown downstream task. In the causal view, each augmented view of a data point has its own proxy label, where both SCL and ACL aim to minimize the differences between the representation outputs of different views of the same data.

and common corruptions [26]. ACL is attracting increasing popularity since the adversarial robustness of the pre-trained models is essential to safety-critical applications [6, 33]. Many studies have tried to improve ACL from various perspectives including increasing contrastive views and leveraging pseudo labels [22], leveraging hard negative sampling [42, 50], and dynamically scheduling the strength of data augmentations [36].

The style-independence property of learned representations, which eliminates the effects of nuisance style factors in SCL, has been shown to improve the transferability of representations [38]. To achieve style-independence property in learned representations, Mitrovic et al. [38] proposed a standard invariant regularization (SIR) that uses causal reasoning [39, 40] to enforce the representations of natural data to be invariant to style factors. SIR has been shown effective in enhancing representation' transferability. However, it remains unclear in the literature how to achieve style independence in robust representations learned via ACL, especially when adversarial data is used during pre-training.

Therefore, we leverage the technique of causal reasoning to enforce robust representations learned via ACL to be style-independent. As shown in the right panel of Figure 1, we construct the causal graph of ACL (details refer to Section 3.1). Different from the causal graph of SCL [38], ACL has an extra path $x \to \tilde{x}$ since ACL will generate the adversarial data $\tilde{x}$ given the unlabeled data $x$ (i.e., $x \to \tilde{x}$). Then, ACL learns representations by maximizing both the probability of the proxy label $y^R$ given the natural data and that given the adversarial data (i.e., $x \to y^R$ and $\tilde{x} \to y^R$). Theorem 1 shows that maximizing the aforementioned probability in the causal view is equivalent to the learning objective of ACL [29], which justifies the rationality of our constructed causal graph of ACL.

To enforce robust representations to be style-independent, we propose an adversarial invariant regularization (AIR). Specifically, AIR (see Eq. (7)) aims to penalize the Kullback–Leibler divergence between the robust representations of the unlabeled data augmented via two different data augmentation functions during the learning procedure of $x \to \tilde{x} \to y^R$. Note that SIR [38] is a special case of AIR in the standard context where $x = \tilde{x}$. Then, we propose to learn robust representations by minimizing the adversarial contrastive loss [29, 22, 50, 36] together with a weighted sum of SIR and AIR and show the learning algorithm in Algorithm 1.

Furthermore, we give a theoretical understanding of the proposed AIR and show that AIR is beneficial to the robustness against corruptions on downstream tasks. Based on the decomposition of AIR shown in Lemma 2, we propose Proposition 3 which explains that AIR implicitly enforces the representational distance between original as well as augmented views of natural data and their adversarial variants to be independent of the style factors. In addition, we theoretically show that the style-independence property of robust representations learned via ACL will still hold on the downstream classification tasks in Proposition 4, which could be helpful to improve robustness against input perturbations on downstream tasks [3, 45, 28, 38].

Empirically, we conducted comprehensive experiments on various datasets including CIFAR-10 [31], CIFAR-100 [31], STL-10 [12], CIFAR-10-C [26], and CIFAR-100-C [26] to show the effectiveness of our proposed method in improving ACL methods [29, 22, 50, 36]. We demonstrate that our proposed method can achieve the new state-of-the-art (SOTA) performance on various downstream tasks by significantly enhancing standard generalization and robustness against adversarial attacks [13, 14, 2] and common corruptions [26]. Notably, compared with the prior SOTA method DynACL [36], AIR improves the standard and robust test accuracy by 2.71% (from 69.59±0.08% to 72.30±0.10%) and 1.17% (from 46.49±0.05% to 47.66±0.06%) on the STL-10 task and increases the test accuracy under common corruptions by 1.15% (from 65.60±0.18% to 66.75±0.10%) on CIFAR-10-C.

## 2 Related Works and Preliminaries

Here, we introduce the related works in ACL and causal reasoning and provide the preliminaries.

### 2.1 Related Works

**Adversarial contrastive learning (ACL).**  Contrastive learning (CL) is frequently used to leverage large unlabeled datasets for learning useful representations. Chen et al. [9] presented SimCLR that leverages contrastive loss for learning useful representations and achieved significantly improved standard test accuracy on downstream tasks. Recently, adversarial contrastive learning (ACL) [30, 29, 27, 22, 50, 52, 36] has become the most effective unsupervised approaches to learn robust representations. Jiang et al. [29], which is the seminal work of ACL that incorporates adversarial data [37] with contrastive loss [9], showed that ACL can exhibit better robustness against adversarial attacks [23, 13] and common corruptions [26] on downstream tasks compared with SCL [9].

One research line focuses on improving the performance of ACL. AdvCL [22] leverages a third contrastive view for high-frequency components and pseudo labels generated by clustering to improve ACL. Yu et al. [50] proposed the asymmetric InfoNCE objective (A-InfoNCE) that utilizes the hard negative sampling method [42] to further enhance AdvCL [22]. Recently, DynACL [36] dynamically schedules the strength of data augmentations and correspondingly schedules the weight for standard and adversarial contrastive losses, achieving the SOTA performance among existing ACL methods. Another research line focuses on improving the efficiency of ACL since ACL requires tremendous running time to generate the adversarial training data during pre-training. Xu et al. [48] proposed a robustness-aware coreset selection (RCS) method to speed up ACL by decreasing the number of training data. Our paper, belonging to the former research line, leveraged the technique of causal reasoning to further enhance ACL and achieved new SOTA results. We treat the incorporation of our proposed method with RCS [48] as future work.

**Causal reasoning.**  Causal reasoning [39, 53, 40, 8, 56, 38, 7, 44, 43] has been widely applied to machine learning to identify causal relations and ignore nuisance factors by intervention. In particular, Zhang et al. [53, 56] investigated the causality in supervised adversarial training [37, 55] where label information is required and proposed the regularizer to eliminate the difference between the natural and adversarial distributions. Mitrovic et al. [38] built the causal graph of SCL in the standard context and introduced the standard invariant regularization (SIR) which aims to enforce the representation of unlabeled natural data to be invariant of the data augmentations. SIR can significantly improve robustness transferability to downstream tasks in terms of common corruptions [26], which is empirically validated by Mitrovic et al. [38].

### 2.2 Preliminaries

**Standard contrastive learning (SCL) [9].**  Let $h_\theta : \mathcal{X} \to \mathcal{Z}$ be a representation extractor parameterized by $\theta$, $g : \mathcal{Z} \to \mathcal{V}$ be a projection head that maps representations to the space where the contrastive loss is applied, and $\tau_i, \tau_j : \mathcal{X} \to \mathcal{X}$ be two transformation operations randomly sampled from a pre-defined transformation set $\mathcal{T}$. Given a minibatch $B \sim \mathcal{X}^\beta$ consisting of $\beta$ original samples, we denote the augmented minibatch $B^u = \{\tau_u(x_k) \mid \forall x_k \in B\}$ via augmentation function $\tau_u(\cdot)$. We take $f_\theta(\cdot) = g \circ h_\theta(\cdot)$ and $x_k^u = \tau_u(x_k)$ for any $x_k \sim \mathcal{X}$ and $u \in \{i, j\}$. Given a positive pair $(x_k^i, x_k^j)$, the standard contrastive loss proposed by SimCLR [9] is formulated as follows:

$$\ell_{\mathrm{CL}}(x_k^i, x_k^j; \theta) = - \sum_{u \in \{i,j\}} \log \frac{e^{\mathrm{sim}\left(f_\theta(x_k^i), f_\theta(x_k^j)\right)/t}}{\sum\limits_{x \in B^i \cup B^j \setminus \{x_k^u\}} e^{\mathrm{sim}\left(f_\theta(x_k^u), f_\theta(x)\right)/t}}, \tag{1}$$

where $\text{sim}(p, q) = p^\top q/\|p\|\|q\|$ is the cosine similarity function and $t > 0$ is a temperature parameter.

**ACL [29] and DynACL [36].**  Let $(\mathcal{X}, d_\infty)$ be the input space $\mathcal{X}$ with the infinity distance metric $d_\infty(x, x') = \|x - x'\|_\infty$, and $\mathcal{B}_\epsilon[x] = \{x' \in \mathcal{X} \mid d_\infty(x, x') \leq \epsilon\}$ be the closed ball of radius $\epsilon > 0$ centered at $x \in \mathcal{X}$. Given a data point $x_k \in \mathcal{X}$, the adversarial contrastive loss is as follows:

$$\ell_{\text{ACL}}(x_k; \theta) = (1 + \omega) \cdot \ell_{\text{CL}}(\tilde{x}_k^i, \tilde{x}_k^j; \theta) + (1 - \omega) \cdot \ell_{\text{CL}}(x_k^i, x_k^j; \theta), \tag{2}$$

$$\text{where} \quad \tilde{x}_k^i, \tilde{x}_k^j = \underset{\substack{\tilde{x}_k^i \in \mathcal{B}_\epsilon[x_k^i] \\ \tilde{x}_k^j \in \mathcal{B}_\epsilon[x_k^j]}}{\arg\max} \ell_{\text{CL}}(\tilde{x}_k^i, \tilde{x}_k^j; \theta), \tag{3}$$

in which $\omega \in [0, 1]$ is a hyperparameter and $\tilde{x}_k^i$ and $\tilde{x}_k^j$ are adversarial data generated via projected gradient descent (PGD) [37, 54, 55] within the $\epsilon$-balls centered at $x_k^i$ and $x_k^j$. Note that ACL [29] fixes $\omega = 0$ while DynACL [36] dynamically schedules $\omega$ according to its dynamic augmentation scheduler that gradually anneals from a strong augmentation to a weak one. We leave the details of the data augmentation scheduler [36] in Appendix C due to the page limitation.

## 3   Methodology

In this section, we first create the causal graph for ACL which is the fundamental premise for causal reasoning. Then, we introduce adversarial invariant regularization (AIR) according to the causal understanding in the context of ACL and the learning algorithm of our proposed method. Lastly, we demonstrate the theoretical understanding of AIR and theoretically show that the style-independence property is generalizable to downstream tasks.

### 3.1   Adversarial Contrastive Learning (ACL) in the View of Causality

A causal graph is arguably the fundamental premise for causal reasoning [39, 40]. Previously, Mitrovic et al. [38] constructed the causal graph of SCL as shown in the left panel of Figure 1. However, how to build causal graphs of ACL is still unclear in the literature. Therefore, we take the primitive step to construct the causal graph of ACL.

To begin with, we formalize the data generation procedure in the causal graph. Let $x \in \mathcal{X}$ denote an unlabeled data point and $\mathcal{Y} = \{y_t\}_{t=1}^T$ denote a set of labels in an unknown downstream task. Then, similar to Mitrovic et al. [38] and Zhang et al. [56] we make the following assumptions: 1) the data point $x$ is generated from the content factor $c$ and the style factor $s$, i.e., $c \to x \leftarrow s$, 2) the content is independent of the style, i.e., $c \perp\!\!\!\perp s$, and 3) only the content, which is the original image data, is related to the downstream task, i.e., $c \to y_t$.

Then, we build the learning procedure of CL. According to the above assumptions, the content serves an essential role in downstream classification tasks. Therefore, CL can be regarded as estimating the probability of the label given the content, i.e., $p(y_t|c)$. To learn the representations from the unlabeled data $x$, CL needs to maximize the conditional probability $p(y_t|x)$.

However, in practice, the label $y_t$ is unknown. Thanks to the causal concept of refinement [8, 38], we can construct a proxy label $y^R \in \mathcal{Y}^R$ which is a refinement of $\mathcal{Y}$. This proxy label $y^R$ should be consistent across different views of the unlabeled data $x$, which enables learning the representations. Intuitively, the proxy label $y^R$ can be viewed as a more fine-grained variant of the original label $y_t$. For example, a refinement for recognizing cars would be the task of classifying various types of cars. Therefore, SCL targets maximizing the probability of the proxy label given the unlabeled natural data, i.e., $p(y^R|x)$ [38].

Besides $x \to y^R$, ACL has an extra path $x \to \tilde{x} \to y^R$ since ACL needs to first generate the adversarial data (Eq. (3)) and then learns representations from the unlabeled natural and adversarial data (Eq. (2)). According to Eq. (3), we observe that ACL only uses the observed unlabeled data $x$ for generating adversarial data, i.e., $x \to \tilde{x} \leftarrow \theta$. Then, according to the paths $x \to y^R$ and $\tilde{x} \to y^R$, from the causal view, ACL learns representations by maximizing both the probability of the proxy label $y^R$ given the natural data (i.e., $p(y^R|x)$) and that given adversarial data (i.e., $p(y^R|\tilde{x})$).

Lastly, we provide Theorem 1 to justify the rationality of our constructed causal graph of ACL.

**Theorem 1.** *From the causal view, in ACL, maximizing the conditional probability both $p(y^R|x)$ and $p(y^R|\tilde{x})$ is equivalent to minimizing the learning objective of ACL [29] that is the sum of standard and adversarial contrastive losses.*

The proof of Theorem 1 is in Appendix B.1.

## 3.2 Adversarial Invariant Regularization

According to the independence of causal mechanisms [40], performing interventions on the style variable $s$ should not change the probability of the proxy label given the unlabeled data, i.e., $p(y^R|x)$. According to the path $x \to \tilde{x} \to y^R$ shown in the causal graph of ACL, we can obtain that

$$p(y^R|x) = p(y^R|\tilde{x})p(\tilde{x}|x), \tag{4}$$

under the assumption that the process of $x \to \tilde{x} \to y^R$ satisfies the Markov condition. Therefore, we should make the conditional probability $p(y^R|\tilde{x})p(\tilde{x}|x)$ that is learned via ACL become style-independent. It guides us to propose the following criterion that should be satisfied by ACL:

$$p^{do(\tau_i)}(y^R|\tilde{x})p^{do(\tau_i)}(\tilde{x}|x) = p^{do(\tau_j)}(y^R|\tilde{x})p^{do(\tau_j)}(\tilde{x}|x) \quad \forall \tau_i, \tau_j \in \mathcal{T}, \tag{5}$$

where $do(\tau)$ as the short form of $do(s = \tau)$ denotes that we perform the intervention on $s$ via data augmentation function $\tau(\cdot)$. We leverage the representational distance between various views of natural data and their adversarial variants normalized by the softmax function to calculate the conditional probability $p^{do(\tau_u)}(y^R|\tilde{x})$ and $p^{do(\tau_u)}(\tilde{x}|x)$ where $u \in \{i, j\}$ as follows:

$$p^{do(\tau_u)}(y^R|\tilde{x}) = \frac{e^{\text{sim}(f_\theta(x), f_\theta(\tilde{x}^u))/t}}{\sum\limits_{x_k \in B} e^{\text{sim}(f_\theta(x_k), f_\theta(\tilde{x}_k^u))/t}}, \quad p^{do(\tau_u)}(\tilde{x}|x) = \frac{e^{\text{sim}(f_\theta(\tilde{x}^u), f_\theta(x^u))/t}}{\sum\limits_{x_k \in B} e^{\text{sim}(f_\theta(\tilde{x}_k^u), f_\theta(x_k^u))/t}}, \tag{6}$$

in which $x \in B \sim \mathcal{X}^\beta$ is the original view of natural data, $x^u$ is the augmented view of natural data via augmentation $\tau_u(\cdot)$, $\tilde{x}^u$ is the adversarial variant generated via Eq. (3), and $t$ is the temperature parameter. Note that we use $x$ as the approximated surrogate of its true content $c$ that incurs $y^R$ (i.e., $c \to y^R$).

To achieve the above criterion in Eq. (5), we propose an adversarial invariant regularization (AIR) to regulate robust representations as follows:

$$\mathcal{L}_{\text{AIR}}(B; \theta, \epsilon) = \text{KL}\left(p^{do(\tau_i)}(y^R|\tilde{x})p^{do(\tau_i)}(\tilde{x}|x)\|p^{do(\tau_j)}(y^R|\tilde{x})p^{do(\tau_j)}(\tilde{x}|x); B\right), \tag{7}$$

in which $\epsilon \geq 0$ is the adversarial budget and $\text{KL}(p(x)\|q(x); B) = \sum_{x \in B} p(x) \log \frac{p(x)}{q(x)}$ denotes the Kullback–Leibler (KL) divergence. AIR can enforce the representation to satisfy the criterion in Eq. (5), thus eliminating the effect of the style factors on the representation. Therefore, when setting $\epsilon$ as a positive constant, $\mathcal{L}_{\text{AIR}}(B; \theta, \epsilon)$ can effectively regulate robust representations of adversarial data to be style-independent.

Besides, to explicitly regulate standard representations of natural data to be independent of style factors, we can simply set $\epsilon = 0$ of AIR. We formulate AIR with $\epsilon = 0$ as follows:

$$\mathcal{L}_{\text{AIR}}(B; \theta, 0) = \text{KL}\left(p^{do(\tau_i)}(y^R|x)\|p^{do(\tau_j)}(y^R|x); B\right), \tag{8}$$

$$\text{where} \quad p^{do(\tau_u)}(y^R|x) = \frac{e^{\text{sim}(f_\theta(x), f_\theta(x^u))/t}}{\sum\limits_{x_k \in B} e^{\text{sim}(f_\theta(x_k), f_\theta(x_k^u))/t}} \quad \forall u \in \{i, j\}.$$

Note that $\mathcal{L}_{\text{AIR}}(B; \theta, 0)$ is the same formulation as the standard invariant regularization (SIR) [38], which can make the standard representation style-independent.

By incorporating adversarial contrastive loss with AIR, the learning objective function of our proposed method is shown as follows:

$$\arg\min_\theta \sum_{x \in U} \ell_{\text{ACL}}(x; \theta) + \lambda_1 \cdot \mathcal{L}_{\text{AIR}}(U; \theta, 0) + \lambda_2 \cdot \mathcal{L}_{\text{AIR}}(U; \theta, \epsilon) \tag{9}$$

where $U \sim \mathcal{X}^N$ refers to an unlabeled dataset consisting of $N$ samples, $\epsilon > 0$ is the adversarial budget, and $\lambda_1 \geq 0$ and $\lambda_2 \geq 0$ are two hyperparameters. The learning algorithm of ACL with AIR is demonstrated in Algorithm 1. Note that our proposed AIR is compatible with various learning objectives such as ACL [29], AdvCL [22], A-InfoNCE [50], and DynACL [36].

**Algorithm 1** ACL with Adversarial Invariant Regularization (AIR)

---
1: **Input:** Unlabeled training set $U$, total training epochs $E$, learning rate $\eta$, batch size $\beta$, adversarial budget $\epsilon > 0$, hyperparameters $\lambda_1$ and $\lambda_2$
2: **Output:** Pre-trained representation extractor $h_\theta$
3: Initialize parameters of model $f_\theta = g \circ h_\theta$
4: **for** $e = 0$ **to** $E - 1$ **do**
5:     **for** batch $m = 1, \ldots, \lceil |U|/\beta \rceil$ **do**
6:         Sample a minibatch $B_m$ from $U$
7:         Update $\theta \leftarrow \theta - \eta \cdot \nabla_\theta \sum_{x_k \in B_m} \ell_{\text{ACL}}(x_k; \theta) + \lambda_1 \cdot \mathcal{L}_{\text{AIR}}(B_m; \theta, 0) + \lambda_2 \cdot \mathcal{L}_{\text{AIR}}(B_m; \theta, \epsilon)$
8:     **end for**
9: **end for**

---

## 3.3 Theoretical Analysis

We provide a theoretical understanding of AIR in Proposition 3 based on the decomposition of AIR shown in Lemma 2. We set $\epsilon > 0$ of AIR by default in this section.

**Lemma 2** (Decomposition of AIR). *AIR in Eq.* (7) *can be decomposed into two terms as follows:*

$$\mathcal{L}_{\text{AIR}}(B; \theta, \epsilon) = \mathbb{E}_{x \sim p^{do(\tau_i)}(\tilde{x}|x)}[\text{KL}(p^{do(\tau_i)}(y^R|\tilde{x}) \| p^{do(\tau_j)}(y^R|\tilde{x}))]$$
$$+ \mathbb{E}_{x \sim p^{do(\tau_i)}(y^R|\tilde{x})}[\text{KL}(p^{do(\tau_i)}(\tilde{x}|x) \| p^{do(\tau_j)}(\tilde{x}|x))],$$

*where $\mathbb{E}_{x \sim Q^3(x)}[\text{KL}(Q^1(x) \| Q^2(x))]$ is the expectation of KL divergence over $Q^3$ and $Q^1, Q^2, Q^3$ are probability distributions.*

The proof of Lemma 2 is in Appendix B.2. Based on Lemma 2, we propose Proposition 3 that provides the theoretical understanding of AIR.

**Proposition 3.** *AIR implicitly enforces the robust representation to satisfy the following two proxy criteria:*

$$(1) \quad p^{do(\tau_i)}(y^R|\tilde{x}) = p^{do(\tau_j)}(y^R|\tilde{x}), \quad (2) \quad p^{do(\tau_i)}(\tilde{x}|x) = p^{do(\tau_j)}(\tilde{x}|x).$$

The proof of Proposition 3 is in Appendix B.3.

**Remarks.** Proposition 3 explains that AIR implicitly enforces the representational distance to be style-independent between the original view of natural data and their adversarial variants (i.e., $p^{do(\tau_i)}(y^R|\tilde{x}) = p^{do(\tau_j)}(y^R|\tilde{x})$), as well as between the augmented view of natural data and their adversarial counterparts (i.e., $p^{do(\tau_i)}(\tilde{x}|x) = p^{do(\tau_j)}(\tilde{x}|x)$). Therefore, Proposition 3 explicitly presents that AIR is different from SIR [38], where SIR only enforces the augmented views of natural data to be style-independent.

Next, we theoretically show that on the assumption that $\mathcal{Y}^R$ is a refinement of $\mathcal{Y}$, the style-independence property of robust representations will hold on downstream classification tasks in Proposition 4.

**Proposition 4.** *Let $\mathcal{Y} = \{y_t\}_{t=1}^T$ be a label set of a downstream classification task, $\mathcal{Y}^R$ be a refinement of $\mathcal{Y}$, and $\tilde{x}_t$ be the adversarial data generated on the downstream task. Assuming that $\tilde{x} \in \mathcal{B}_\epsilon[x]$ and $\tilde{x}_t \in \mathcal{B}_\epsilon[x]$, we have the following results:*

$$p^{do(\tau_i)}(y^R|\tilde{x}) = p^{do(\tau_j)}(y^R|\tilde{x}) \Longrightarrow p^{do(\tau_i)}(y_t|\tilde{x}_t) = p^{do(\tau_j)}(y_t|\tilde{x}_t) \quad \forall \tau_i, \tau_j \in \mathcal{T},$$
$$p^{do(\tau_i)}(\tilde{x}|x) = p^{do(\tau_j)}(\tilde{x}|x) \Longrightarrow p^{do(\tau_i)}(\tilde{x}_t|x) = p^{do(\tau_j)}(\tilde{x}_t|x) \quad \forall \tau_i, \tau_j \in \mathcal{T}.$$

The proof of Proposition 4 is shown in Appendix B.4.

**Remarks.** Proposition 4 indicates that the robust representation of data on downstream tasks is still style-independent. Mitrovic et al. [38] empirically showed that the style-independence property of standard representations can significantly improve the robustness against common corruptions [26] on downstream tasks. In addition, Proposition 4 indicates that AIR helps to find the invariant correlations that are independent of the style factors among different training distributions. It is similar to the objective of invariant risk minimization [3] that can yield substantial improvement in the robustness against various corruptions incurred by distribution shifts. Therefore, the style-independence property of robust representations learned via ACL perhaps helps to enhance the robustness against input perturbations on downstream tasks.

Table 1: Self-task adversarial robustness transferability.

| Dataset | Pre-training | SLF | | ALF | | AFF | |
|---|---|---|---|---|---|---|---|
| | | AA (%) | SA (%) | AA (%) | SA (%) | AA (%) | SA (%) |
| CIFAR-10 | ACL [29] | $37.39_{\pm0.06}$ | $78.27_{\pm0.09}$ | $40.61_{\pm0.07}$ | $75.56_{\pm0.09}$ | $49.42_{\pm0.07}$ | $82.14_{\pm0.18}$ |
| | ACL-AIR | $\mathbf{38.89}_{\pm0.06}$ | $\mathbf{80.03}_{\pm0.07}$ | $\mathbf{41.39}_{\pm0.08}$ | $\mathbf{78.29}_{\pm0.10}$ | $\mathbf{49.84}_{\pm0.04}$ | $\mathbf{82.42}_{\pm0.06}$ |
| | DynACL [36] | $45.05_{\pm0.04}$ | $75.39_{\pm0.05}$ | $45.65_{\pm0.05}$ | $72.90_{\pm0.08}$ | $50.52_{\pm0.05}$ | $81.86_{\pm0.11}$ |
| | DynACL-AIR | $\mathbf{45.17}_{\pm0.04}$ | $\mathbf{78.08}_{\pm0.06}$ | $\mathbf{46.01}_{\pm0.07}$ | $\mathbf{77.42}_{\pm0.10}$ | $\mathbf{50.60}_{\pm0.08}$ | $\mathbf{82.14}_{\pm0.11}$ |
| CIFAR-100 | ACL [29] | $15.78_{\pm0.05}$ | $45.70_{\pm0.10}$ | $17.36_{\pm0.16}$ | $42.69_{\pm0.13}$ | $24.16_{\pm0.29}$ | $56.68_{\pm0.14}$ |
| | ACL-AIR | $\mathbf{16.14}_{\pm0.07}$ | $\mathbf{49.75}_{\pm0.10}$ | $\mathbf{18.68}_{\pm0.11}$ | $\mathbf{47.07}_{\pm0.15}$ | $\mathbf{25.27}_{\pm0.10}$ | $\mathbf{57.79}_{\pm0.18}$ |
| | DynACL [36] | $19.31_{\pm0.06}$ | $45.67_{\pm0.09}$ | $20.30_{\pm0.08}$ | $43.58_{\pm0.12}$ | $24.70_{\pm0.23}$ | $57.22_{\pm0.28}$ |
| | DynACL-AIR | $\mathbf{20.45}_{\pm0.07}$ | $\mathbf{46.84}_{\pm0.12}$ | $\mathbf{21.23}_{\pm0.09}$ | $\mathbf{45.63}_{\pm0.10}$ | $\mathbf{25.34}_{\pm0.12}$ | $\mathbf{57.44}_{\pm0.14}$ |
| STL-10 | ACL [29] | $35.80_{\pm0.06}$ | $67.90_{\pm0.09}$ | $38.10_{\pm0.11}$ | $69.96_{\pm0.14}$ | $43.21_{\pm0.16}$ | $72.55_{\pm0.18}$ |
| | ACL-AIR | $\mathbf{36.94}_{\pm0.06}$ | $\mathbf{68.91}_{\pm0.07}$ | $\mathbf{39.05}_{\pm0.10}$ | $\mathbf{71.30}_{\pm0.12}$ | $\mathbf{43.75}_{\pm0.13}$ | $\mathbf{72.84}_{\pm0.14}$ |
| | DynACL [36] | $46.49_{\pm0.05}$ | $69.59_{\pm0.08}$ | $47.69_{\pm0.10}$ | $67.65_{\pm0.12}$ | $45.64_{\pm0.13}$ | $72.14_{\pm0.15}$ |
| | DynACL-AIR | $\mathbf{47.66}_{\pm0.06}$ | $\mathbf{72.30}_{\pm0.10}$ | $\mathbf{48.89}_{\pm0.08}$ | $\mathbf{71.60}_{\pm0.09}$ | $\mathbf{48.10}_{\pm0.11}$ | $\mathbf{73.10}_{\pm0.17}$ |

Table 2: Self-task common-corruption robustness transferability. The corruption severity (CS) ranging from $\{1,3,5\}$ (denoted as "CS-$\{1,3,5\}$"). The standard deviation is in Table 15.

| Dataset | Pre-training | SLF | | | ALF | | | AFF | | |
|---|---|---|---|---|---|---|---|---|---|---|
| | | CS-1 | CS-3 | CS-5 | CS-1 | CS-3 | CS-5 | CS-1 | CS-3 | CS-5 |
| CIFAR-10 | ACL [29] | 76.57 | 71.78 | 62.78 | 74.04 | 69.49 | 61.38 | 79.15 | 72.54 | 65.27 |
| | ACL-AIR | **78.55** | **73.33** | **64.28** | **76.65** | **71.38** | **63.17** | **79.49** | **72.95** | **65.37** |
| | DynACL [36] | 73.92 | 69.01 | 62.51 | 71.74 | 66.95 | 60.87 | 79.77 | 72.95 | 65.60 |
| | DynACL-AIR | **76.62** | **70.16** | **63.29** | **75.70** | **69.55** | **62.67** | **80.98** | **74.31** | **66.75** |
| CIFAR-100 | ACL [29] | 43.14 | 38.90 | 32.27 | 39.52 | 35.13 | 29.30 | 53.80 | 45.93 | 37.61 |
| | ACL-AIR | **47.85** | **41.54** | **34.12** | **44.80** | **40.70** | **35.46** | **55.12** | **47.25** | **39.02** |
| | DynACL [36] | 44.19 | 38.08 | 31.05 | 39.83 | 35.61 | 30.51 | 54.34 | 46.71 | 38.62 |
| | DynACL-AIR | **45.36** | **39.21** | **32.44** | **42.03** | **36.48** | **30.49** | **55.31** | **47.33** | **39.13** |

## 4 Experiments

In this section, we demonstrate the effectiveness of our proposed AIR in improving ACL [29] and its variants [36, 22, 50] on various datasets including CIFAR-10 [31], CIFAR-100 [31], STL-10 [12], CIFAR-10-C [26], and CIFAR-100-C [26]. Extensive experimental details are shown in Appendix C.

**Pre-training.** In the main paper, we demonstrate the results of applying our proposed AIR with $\lambda_1 = 0.5$ and $\lambda_2 = 0.5$ to ACL [29] and DynACL [36] (denoted as "ACL-AIR" and "DynACL-AIR", respectively). We utilized ResNet-18 [25] as the representation extractor following previous self-supervised adversarial training methods [29, 22, 36, 50]. We adopted the same training configuration of ACL [29] using SGD for 1000 epochs with an initial learning rate of 5.0 and a cosine annealing schedule [35]. The batch size $\beta$ is fixed as 512. The adversarial budget $\epsilon$ is set as $8/255$. In the context of DynACL, we took the same data augmentation scheduler and the same scheduler of the hyperparameter $\omega$ as the setting in the original paper of DynACL [36]. Note that, to reproduce the results of baseline methods, we downloaded the pre-trained weights of ACL [29] on CIFAR-10/CIFAR-100 and DynACL [36] on CIFAR-10/CIFAR-100/STL-10 from their official GitHub as the pre-trained representation extractor. We provide the extensive ablation studies on the hyper-parameters ($\lambda_1$ and $\lambda_2$) in Appendix C.2, the ACL variants (including AdvCL [22] and A-InfoNCE [50]) in Appendix C.3, and the backbone models (including ResNet-50 and Wide ResNet [51]) in Appendix C.4.

**Finetuning procedures.** We adopted the following three types of finetuning procedures: standard linear finetuning (SLF), adversarial linear finetuning (ALF), and adversarial full finetuning (AFF), to evaluate the learned representations. The former two finetuning procedures freeze the learned representation extractor and only train the linear classifier using natural or adversarial samples, respectively. We took the pre-trained representation extractor as weight initialization and trained the whole model using the adversarial data during AFF. The training configuration of finetuning (e.g., the finetuning epoch and optimizer) exactly follows DynACL [36]. Specifically, we used the official code provided in DynACL [36]'s GitHub for finetuning and illustrate the experimental settings of finetuning in Appendix C as well. For each pre-trained representation extractor, we repeated the finetuning experiments 3 times and report the median results and the standard deviation.

**Evaluation protocols.** We let "AA" denote the robust test accuracy under AutoAttack (AA) [13] and "SA" stand for the standard test accuracy. To evaluate the robustness against common corruption, we report the mean test accuracy under 15 types of common corruptions [26] with the corruption

Table 3: Cross-task adversarial robustness transferability. $\mathcal{D}_1 \to \mathcal{D}_2$ denotes pre-training and finetuning are conducted on the dataset $\mathcal{D}_1$ and $\mathcal{D}_2(\neq \mathcal{D}_1)$, respectively.

| $\mathcal{D}_1 \to \mathcal{D}_2$ | Pre-training | SLF | | ALF | | AFF | |
|---|---|---|---|---|---|---|---|
| | | AA (%) | SA (%) | AA (%) | SA (%) | AA (%) | SA (%) |
| CIFAR-10 $\to$ CIFAR-100 | ACL [29] | $9.98_{\pm 0.02}$ | $32.61_{\pm 0.04}$ | $11.09_{\pm 0.06}$ | $28.58_{\pm 0.06}$ | $22.67_{\pm 0.16}$ | $56.05_{\pm 0.19}$ |
| | ACL-AIR | $\mathbf{11.04}_{\pm 0.06}$ | $\mathbf{39.45}_{\pm 0.07}$ | $\mathbf{13.30}_{\pm 0.02}$ | $\mathbf{36.10}_{\pm 0.05}$ | $\mathbf{23.45}_{\pm 0.04}$ | $\mathbf{56.31}_{\pm 0.06}$ |
| | DynACL [36] | $11.01_{\pm 0.02}$ | $27.66_{\pm 0.03}$ | $11.92_{\pm 0.05}$ | $24.14_{\pm 0.09}$ | $24.17_{\pm 0.10}$ | $55.61_{\pm 0.17}$ |
| | DynACL-AIR | $\mathbf{12.20}_{\pm 0.04}$ | $\mathbf{31.33}_{\pm 0.03}$ | $\mathbf{12.70}_{\pm 0.05}$ | $\mathbf{28.70}_{\pm 0.05}$ | $\mathbf{24.82}_{\pm 0.07}$ | $\mathbf{57.00}_{\pm 0.13}$ |
| CIFAR-10 $\to$ STL-10 | ACL [29] | $25.41_{\pm 0.08}$ | $56.53_{\pm 0.10}$ | $27.17_{\pm 0.09}$ | $51.71_{\pm 0.17}$ | $32.66_{\pm 0.07}$ | $61.41_{\pm 0.13}$ |
| | ACL-AIR | $\mathbf{28.00}_{\pm 0.12}$ | $\mathbf{61.91}_{\pm 0.13}$ | $\mathbf{30.06}_{\pm 0.10}$ | $\mathbf{62.03}_{\pm 0.11}$ | $\mathbf{34.26}_{\pm 0.09}$ | $\mathbf{62.58}_{\pm 0.10}$ |
| | DynACL [36] | $28.52_{\pm 0.09}$ | $52.45_{\pm 0.10}$ | $29.13_{\pm 0.13}$ | $49.53_{\pm 0.17}$ | $35.25_{\pm 0.15}$ | $63.29_{\pm 0.18}$ |
| | DynACL-AIR | $\mathbf{29.88}_{\pm 0.04}$ | $\mathbf{54.59}_{\pm 0.12}$ | $\mathbf{31.24}_{\pm 0.06}$ | $\mathbf{57.14}_{\pm 0.09}$ | $\mathbf{35.66}_{\pm 0.05}$ | $\mathbf{63.74}_{\pm 0.12}$ |
| CIFAR-100 $\to$ CIFAR-10 | ACL [29] | $18.72_{\pm 0.07}$ | $60.90_{\pm 0.02}$ | $26.92_{\pm 0.11}$ | $57.35_{\pm 0.07}$ | $44.07_{\pm 0.11}$ | $75.19_{\pm 0.10}$ |
| | ACL-AIR | $\mathbf{19.90}_{\pm 0.04}$ | $\mathbf{64.89}_{\pm 0.09}$ | $\mathbf{27.65}_{\pm 0.06}$ | $\mathbf{60.79}_{\pm 0.04}$ | $\mathbf{44.84}_{\pm 0.14}$ | $\mathbf{75.67}_{\pm 0.13}$ |
| | DynACL [36] | $25.23_{\pm 0.12}$ | $59.12_{\pm 0.10}$ | $28.92_{\pm 0.10}$ | $56.09_{\pm 0.14}$ | $47.40_{\pm 0.23}$ | $77.92_{\pm 0.18}$ |
| | DynACL-AIR | $\mathbf{25.63}_{\pm 0.07}$ | $\mathbf{59.83}_{\pm 0.08}$ | $\mathbf{29.32}_{\pm 0.06}$ | $\mathbf{56.65}_{\pm 0.06}$ | $\mathbf{47.92}_{\pm 0.12}$ | $\mathbf{78.44}_{\pm 0.10}$ |
| CIFAR-100 $\to$ STL-10 | ACL [29] | $21.77_{\pm 0.07}$ | $46.19_{\pm 0.05}$ | $24.46_{\pm 0.09}$ | $45.40_{\pm 0.12}$ | $28.76_{\pm 0.07}$ | $56.16_{\pm 0.13}$ |
| | ACL-AIR | $\mathbf{22.44}_{\pm 0.04}$ | $\mathbf{51.52}_{\pm 0.02}$ | $\mathbf{26.55}_{\pm 0.06}$ | $\mathbf{53.24}_{\pm 0.09}$ | $\mathbf{30.40}_{\pm 0.08}$ | $\mathbf{58.45}_{\pm 0.11}$ |
| | DynACL [36] | $23.17_{\pm 0.09}$ | $47.54_{\pm 0.14}$ | $26.24_{\pm 0.13}$ | $45.70_{\pm 0.14}$ | $31.17_{\pm 0.14}$ | $58.35_{\pm 0.18}$ |
| | DynACL-AIR | $\mathbf{23.24}_{\pm 0.07}$ | $\mathbf{48.20}_{\pm 0.08}$ | $\mathbf{26.60}_{\pm 0.05}$ | $\mathbf{48.55}_{\pm 0.12}$ | $\mathbf{31.42}_{\pm 0.07}$ | $\mathbf{58.59}_{\pm 0.10}$ |

Table 4: Cross-task common-corruption robustness transferability. The corruption severity (CS) ranging from $\{1, 3, 5\}$ (denoted as "CS-$\{1,3,5\}$"). The standard deviation is in Table 16.

| $\mathcal{D}_1 \to \mathcal{D}_2$ | Pre-training | SLF | | | ALF | | | AFF | | |
|---|---|---|---|---|---|---|---|---|---|---|
| | | CS-1 | CS-3 | CS-5 | CS-1 | CS-3 | CS-5 | CS-1 | CS-3 | CS-5 |
| CIFAR-10 $\to$ CIFAR-100 | ACL [29] | 31.39 | 27.76 | 23.27 | 27.80 | 25.09 | 21.09 | 52.07 | 44.22 | 36.31 |
| | ACL-AIR | **37.82** | **32.83** | **27.06** | **34.67** | **30.09** | **25.07** | **52.81** | **44.95** | **37.01** |
| | DynACL [36] | 26.74 | 23.97 | 20.87 | 23.70 | 21.43 | 18.84 | 52.87 | 44.87 | 36.76 |
| | DynACL-AIR | **30.35** | **26.53** | **22.76** | **27.67** | **24.35** | **20.96** | **54.00** | **46.01** | **37.75** |
| CIFAR-100 $\to$ CIFAR-10 | ACL [29] | 59.65 | 55.14 | 49.09 | 56.15 | 51.96 | 47.04 | 72.94 | 66.05 | 59.17 |
| | ACL-AIR | **63.34** | **58.14** | **51.03** | **59.14** | **54.23** | **48.97** | **73.48** | **66.83** | **60.21** |
| | DynACL [36] | 57.14 | 52.07 | 47.03 | 56.31 | 52.03 | 47.40 | 75.89 | 69.08 | 62.02 |
| | DynACL-AIR | **58.41** | **53.09** | **47.94** | **54.89** | **49.84** | **45.08** | **76.35** | **69.49** | **62.44** |

severity (CS) ranging from $\{1, 3, 5\}$ (denoted as "CS-$\{1, 3, 5\}$"). Specifically, we used the official code of AutoAttack [13] and RobustBench [15] for implementing evaluations. In Appendix C.5, we provide robustness evaluation under more diverse attacks [13, 14, 2]. In Appendix C.6, we provide the test accuracy under each type of common corruption [26].

## 4.1 Evaluation of Self-Task Robustness Transferability

**Self-task adversarial robustness transferability.** Table 1 reports the self-task adversarial robustness transferability evaluated on three datasets where pre-training and finetuning are conducted on the same datasets including CIFAR-10, CIFAR-100, and STL-10. In Appendix C.1, we report the self-task robustness transferability evaluated in the semi-supervised settings. Table 1 demonstrates that our proposed AIR can obtain new state-of-the-art (SOTA) both standard and robust test accuracy compared with existing ACL methods [29, 36]. Therefore, It validates the effectiveness of our proposed AIR in consistently improving the self-task adversarial robustness transferability against adversarial attacks as well as standard generalization in downstream datasets via various finetuning procedures. Notably, compared with the previous SOTA method DynACL [36], DynACL-AIR increases standard test accuracy by 4.52% (from 72.90% to 77.42%) on the CIFAR-10 dataset via ALF, 2.05% (from 43.58% to 45.63%) on the CIFAR-100 dataset via ALF, and 1.17% (from 46.49% to 47.66%) on the STL-10 dataset via SLF.

**Self-task common-corruption robustness transferability.** Table 2 reports the self-task common-corruption [26] robustness transferability. Specifically, we conducted pre-training on CIFAR-10/CIFAR-100 and then evaluated the test accuracy on CIFAR-10-C/CIFAR-100-C with various corruption severities after various finetuning procedures. Note that the test accuracy under each type of common corruption is reported in Appendix C.6. Table 2 demonstrates that AIR leads to consistent and significant improvement in the robustness against common corruption. In addition, we observe that ACL always achieves much higher test accuracy under common corruptions than DynACL after SLF. We conjecture the reason is that DynACL uses weak data augmentations at the later phase of training, which makes DynACL more likely to overfit the training distribution of CIFAR-10 and thus leads to worse performance under common corruptions via SLF.

Table 5: Self-task adversarial robustness transferability with post-processing. "++" denotes that the pre-trained models are post-processed via LP-AFF [32, 36]. LP-AFF [32] first generates pseudo labels via clustering and then adversarially pre-trains the model using the pseudo labels.

| Dataset | Pre-training | SLF | | ALF | | AFF | |
|---|---|---|---|---|---|---|---|
| | | AA (%) | SA (%) | AA (%) | SA (%) | AA (%) | SA (%) |
| CIFAR-10 | DynACL++ [36] | 46.46$^\dagger$ | 79.81$^\dagger$ | 47.95$^\dagger$ | 78.84$^\dagger$ | 50.31$^\dagger$ | 81.94$^\dagger$ |
| | DynACL-AIR++ | **46.99** | **81.80** | **48.23** | **79.56** | **50.65** | **82.36** |
| CIFAR-100 | DynACL++ [36] | 20.07$^\dagger$ | 52.26$^\dagger$ | 22.24 | 49.92 | 25.21 | 57.30 |
| | DynACL-AIR++ | **20.61** | **53.93** | **22.96** | **52.09** | **25.48** | **57.57** |
| STL-10 | DynACL++ [36] | 47.21$^\dagger$ | 70.93$^\dagger$ | 48.06 | 69.51 | 41.84 | 72.36 |
| | DynACL-AIR++ | **47.90** | **71.44** | **48.59** | **71.45** | **44.09** | **72.42** |

Table 6: Self-task adversarial robustness transferability evaluated on the Imagenette dataset.

| Pre-training | ResNet-18 | | ResNet-50 | |
|---|---|---|---|---|
| | AA (%) | SA (%) | AA (%) | SA (%) |
| DynACL [36] | 57.15 | 79.41 | 58.98 | 80.74 |
| DynACL-AIR | **58.34** | **80.61** | **60.10** | **81.66** |

## 4.2 Evaluation of Cross-Task Robustness Transferability

**Cross-task adversarial robustness transferability.** Table 3 shows the cross-task adversarial robustness transferability where pre-training and finetuning are conducted on the different datasets. We can observe that AIR substantially improves both ACL's and DynACL's robustness against adversarial attacks [13] and standard generalization to other downstream datasets. Particularly, AIR improves the standard and robust test accuracy of ACL via ALF by 7.52% (from 28.58% to 36.10%) and 2.21% (from 11.09% to 13.30%), respectively.

**Cross-task common-corruption robustness transferability.** Table 4 reports the cross-task common-corruption [26] robustness transferability where pre-training is conducted on CIFAR-10/CIFAR-100 and finetuning is conducted on CIFAR-100-C/CIFAR-10-C [26]. The empirical results show that our proposed AIR can consistently improve the accuracy under common corruptions with various severity, which validates the effectiveness of AIR in enhancing the robustness transferability against common corruptions.

## 4.3 Evaluation of Pre-Trained Models after Post-Processing

Here, we report the self-task adversarial robustness transferability of pre-trained models after post-processing in Table 5. The post-processing method is linear probing and then adversarial full finetuning (LP-AFF) [32] which first generates pseudo labels via clustering and then further adversarially trains the pre-trained model using the pseudo labels. "++" denotes that the pre-trained model is post-processed via LP-AFF. We provide the comparison between DynACL++ and DynACL-AIR++ in Table 5. Note that the superscript † denotes that the results of DynACL++ are copied from Luo et al. [36]. The results validate that AIR is compatible with the trick of post-processing via LP-AFF [32], which can yield improved performance on various downstream datasets.

## 4.4 Evaluation on High-Resolution Imagenette Dataset

To the best of our knowledge, there exists no reported result of ACL and DynACL on high-resolution datasets in the existing papers. Therefore, we pre-trained ResNet-18 and ResNet-50 on Imagenette[3] of resolution $256 \times 256$ which is a subset of ImageNet-1K using DynACL and DynACL-AIR, respectively. We compare and report the performance evaluated on Imagenette via SLF in Table 6. Our results validate that our proposed AIR is effective in enhancing the performance on high-resolution datasets. Note that, due to the limitation of our GPU memory, we set the batch size to 256 and 128 for pre-training ResNet-18 and ResNet-50 on Imagenette, respectively. We believe using a larger batch size during pre-training can even further improve the performance [9].

## 4.5 Evaluation via Automated Robust Fine-Tuning

While evaluating the transferability of a pre-trained model, we observed that the performance on the downstream dataset is sensitive to the hyper-parameters (e.g., the learning rate) of the finetuning

---

[3]We downloaded the Imagenette dataset from https://github.com/fastai/imagenette.

Table 7: Cross-task adversarial robustness transferability evaluated via AutoLoRa [47]. $\mathcal{D}_1 \to \mathcal{D}_2$ denotes pre-training and finetuning are conducted on the dataset $\mathcal{D}_1$ and $\mathcal{D}_2(\neq \mathcal{D}_1)$, respectively. "Diff" refers to the gap between the performance achieved by AutoLoRa and that achieved by vanilla finetuning (reported in Table 3).

| $\mathcal{D}_1 \to \mathcal{D}_2$ | Finetuning mode | Pre-training | AutoLoRa [47] | | Diff | |
|---|---|---|---|---|---|---|
| | | | AA (%) | SA (%) | AA (%) | SA (%) |
| CIFAR-10 $\to$ STL-10 | SLF | DynACL [36] | 29.68 | 58.24 | +0.51 | +5.83 |
| | | DynACL-AIR | **29.75** | **60.53** | **+0.11** | **+4.69** |
| | ALF | DynACL [36] | 31.34 | 57.74 | +1.75 | +8.19 |
| | | DynACL-AIR | **31.65** | **59.44** | **+0.41** | **+2.30** |
| | AFF | DynACL [36] | 35.55 | 64.16 | +0.30 | +0.63 |
| | | DynACL-AIR | **35.81** | **64.28** | **+0.15** | **+0.54** |
| CIFAR-100 $\to$ STL-10 | SLF | DynACL [36] | 23.31 | 50.93 | +0.14 | +3.39 |
| | | DynACL-AIR | **23.49** | **51.28** | **+0.25** | **+3.08** |
| | ALF | DynACL [36] | 26.53 | 51.74 | +0.29 | +6.04 |
| | | DynACL-AIR | **26.89** | **49.02** | **+0.29** | **+0.47** |
| | AFF | DynACL [36] | 31.25 | 58.44 | +0.08 | +0.09 |
| | | DynACL-AIR | **31.57** | **58.65** | **+0.15** | **+0.21** |

procedure. It would require extensive computational resources to search for appropriate hyper-parameters to achieve a satisfactory performance. To mitigate this issue, we leverage an automated robust finetuning framework called AutoLoRa [47], which can automatically schedule the learning rate and set appropriate hyper-parameters during finetuning. We report the performance achieved by AutoLoRa to further justify the SOTA performance by our proposed AIR in Table 7.

Table 7 shows that AutoLoRa can further enhance the performance of a pre-trained model on the downstream tasks without searching for appropriate hyper-parameters since the value of "Diff" is consistently larger than 0.0. Besides, Table 7 justifies that our proposed AIR is effective in enhancing robustness transferability via various finetuning methods.

## 5 Conclusions

This paper leveraged the technique of causal reasoning to interpret ACL and proposed adversarial invariant regularization (AIR) to enhance ACL. AIR can enforce the learned robust representations to be invariant of the style factors. We improved ACL by incorporating the adversarial contrastive loss with a weighted sum of AIR and SIR that is an extension of AIR in the standard context. Theoretically, we showed that AIR implicitly encourages the representational distance between different views of natural data and their adversarial variants to be independent of style factors. Empirically, comprehensive results validate that our proposed method can achieve new state-of-the-art performance in terms of standard generalization and robustness against adversarial attacks and common corruptions on downstream tasks.

## Acknowledgements

This research is supported by the National Research Foundation, Singapore under its Strategic Capability Research Centres Funding Initiative, Australian Research Council (ARC) under Award No. DP230101540 and NSF and CSIRO Responsible AI Program under Award No. 2303037. Any opinions, findings and conclusions or recommendations expressed in this material are those of the author(s) and do not reflect the views of National Research Foundation, Singapore.

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

# A  Mathematical Notations

Table 8: Notation Table

| Notation | Description |
|---|---|
| $\mathcal{X}$ | input space |
| $x \in \mathcal{X}$ | data point |
| $\mathcal{T}$ | transformations set |
| $\tau \in \mathcal{T}$ | data augmentation fraction |
| $x^u$ | augmented data point via the data augmentation fraction $\tau_u(\cdot)$ |
| $\tilde{x}$ | adversarial data |
| $B \sim \mathcal{X}^\beta$ | a minibatch of $\beta$ original image samples |
| $B^u$ | a minibatch of $\beta$ augmented image samples via the data augmentation fraction $\tau_u(\cdot)$ |
| $\tilde{B}^u$ | a minibatch of adversarial counterparts of $\beta$ augmented samples via the data augmentation fraction $\tau_u(\cdot)$ |
| $U \sim \mathcal{X}^N$ | an unlabeled data set of $N$ image samples |
| $\mathcal{Y}$ | label space on the downstream tasks |
| $y_t \in \mathcal{Y}$ | target label on the downstream tasks |
| $\mathcal{Y}^R$ | the refinement of the target label space |
| $y^R \in \mathcal{Y}^R$ | the proxy label |
| $y_{ku}^R \in \mathcal{Y}^R$ | the proxy label of the data point $x_k^u$ augmented via $\tau_u(\cdot)$ |
| $\mathcal{Z}$ | projection space |
| $\mathcal{V}$ | feature space where the contrastive loss is applied |
| $h_\theta : \mathcal{X} \to \mathcal{Z}$ | the representation extractor parameterized by $\theta$ |
| $g : \mathcal{Z} \to \mathcal{V}$ | the projector |
| $f(\cdot) = g \circ h_\theta(\cdot)$ | the composition of the representation extractor and the projector |
| $\text{sim}(\cdot, \cdot)$ | the cosine similarity function |
| $\text{KL}(\cdot \| \cdot)$ | KL divergence function |
| $s$ | style variable |
| $c$ | content variable |
| $\lambda_1$ | the weight of standard regularization |
| $\lambda_2$ | the weight of adversarial regularization |

# B  Proof

## B.1  Proof of Theorem 1

**Theorem 1 (restated).**  *From the causal view, in ACL, maximizing the conditional probability both $p(y^R|x)$ and $p(y^R|\tilde{x})$ is equivalent to minimizing the learning objective of ACL [29] that is the sum of standard and adversarial contrastive losses.*

*Proof.*  To begin with, we formulate the proxy label driven $y^R$ by the data augmentation in the context of contrastive learning. We denote the index of $x_k^i \in B^i$ and $x_k^j \in B^j$ as $ki$ and $kj$, respectively. For the augmented data $x_k^i \in B^i$ and $x_k^j \in B^j$, we denote their corresponding proxy labels as $y_{ki}^R$ and $y_{kj}^R$, which is formulated as follows:

$$y_{ki}^R = \mathbf{1}_{kj}, \quad y_{kj}^R = \mathbf{1}_{ki}, \tag{10}$$

where $\mathbf{1}_{ku} \in \{0, 1\}^{2\beta - 1}$ is a one-hot label where $u \in \{i, j\}$, the value at index $ku$ is 1, and the values at other indexes are 0. The one-hot label refers to the refined proxy label of the augmented view of the data point, which directs to its peer view. In the causal view of SCL and ACL, each augmented view of the data has its own proxy label. Therefore, (A)CL aims to minimize the differences in representation output of different views of the same data. Specifically, the conditional probability $p(y_{ki}^R|x_k^i; \theta)$ and $p(y_{kj}^R|x_k^j; \theta)$ is formulated as follows:

$$p(y_{ki}^R|x_k^i; \theta) = p(\mathbf{1}_{kj}|x_k^i; \theta) = \frac{e^{\text{sim}(f_\theta(x_k^i), f_\theta(x_k^j))/t}}{\sum\limits_{x \in B^i \cup B^j \setminus \{x_k^i\}} e^{\text{sim}(f_\theta(x_k^i), f_\theta(x))/t}}, \tag{11}$$

$$p(y_{kj}^R|x_k^j; \theta) = p(\mathbf{1}_{ki}|x_k^j; \theta) = \frac{e^{\text{sim}(f_\theta(x_k^j), f_\theta(x_k^i))/t}}{\sum\limits_{x \in B^i \cup B^j \setminus \{x_k^j\}} e^{\text{sim}(f_\theta(x_k^j), f_\theta(x))/t}}, \tag{12}$$

where $\text{sim}(p, q) = p^\top q / \|p\|\|q\|$ is the cosine similarity function and $t > 0$ is a temperature parameter.

Then, we have the following results:

$$\arg\min_\theta \sum_{x_k \in U} \ell_{\text{CL}}(x_k^i, x_k^j; \theta) \tag{13}$$

$$= \arg\min_\theta \sum_{x_k \in U} -\log \frac{e^{\text{sim}(f_\theta(x_k^i), f_\theta(x_k^j))/t}}{\sum\limits_{x \in B^i \cup B^j \setminus \{x_k^i\}} e^{\text{sim}(f_\theta(x_k^i), f_\theta(x))/t}} - \log \frac{e^{\text{sim}(f_\theta(x_k^j), f_\theta(x_k^i))/t}}{\sum\limits_{x \in B^i \cup B^j \setminus \{x_k^j\}} e^{\text{sim}(f_\theta(x_k^j), f_\theta(x))/t}} \tag{14}$$

$$= \arg\max_\theta \sum_{x_k \in U} \log \frac{e^{\text{sim}(f_\theta(x_k^i), f_\theta(x_k^j))/t}}{\sum\limits_{x \in B^i \cup B^j \setminus \{x_k^i\}} e^{\text{sim}(f_\theta(x_k^i), f_\theta(x))/t}} + \log \frac{e^{\text{sim}(f_\theta(x_k^j), f_\theta(x_k^i))/t}}{\sum\limits_{x \in B^i \cup B^j \setminus \{x_k^j\}} e^{\text{sim}(f_\theta(x_k^j), f_\theta(x))/t}} \tag{15}$$

$$= \arg\max_\theta \sum_{x_k \in U} \frac{e^{\text{sim}(f_\theta(x_k^i), f_\theta(x_k^j))/t}}{\sum\limits_{x \in B^i \cup B^j \setminus \{x_k^i\}} e^{\text{sim}(f_\theta(x_k^i), f_\theta(x))/t}} + \frac{e^{\text{sim}(f_\theta(x_k^j), f_\theta(x_k^i))/t}}{\sum\limits_{x \in B^i \cup B^j \setminus \{x_k^j\}} e^{\text{sim}(f_\theta(x_k^j), f_\theta(x))/t}} \tag{16}$$

$$= \arg\max_\theta \sum_{x_k \in U} p(\mathbf{1}_{kj}|x_k^i; \theta) + \sum_{x_k \in U} p(\mathbf{1}_{ki}|x_k^j; \theta) \tag{17}$$

$$= \arg\max_\theta \sum_{x_k \in U} p(y_{ki}^R|x_k^i; \theta) + p(y_{kj}^R|x_k^j; \theta). \tag{18}$$

Therefore, we can conclude that SCL actually maximizes the conditional probability of the proxy label given the unlabeled data, i.e., $p(y^R|x)$.

In the adversarial context, we only need to replace the natural data $x_k^i$ and $x_k^j$ in Eq. (13) and (18) with the adversarial data $\tilde{x}_k^i$ and $\tilde{x}_k^j$ generated in Eq. (3). Therefore, we have the following results:

$$\arg\min_\theta \sum_{x_k \in U} \ell_{\text{CL}}(x_k^i, x_k^j; \theta) + \ell_{\text{CL}}(\tilde{x}_k^i, \tilde{x}_k^j; \theta)$$
$$= \arg\max_\theta \sum_{x_k \in U} p(y_{ki}^R|x_k^i; \theta) + p(y_{kj}^R|x_k^j; \theta) + p(y_{ki}^R|\tilde{x}_k^i; \theta) + p(y_{kj}^R|\tilde{x}_k^j; \theta), \tag{19}$$

where the adversarial data $\tilde{x}_k^i$ and $\tilde{x}_k^j$ are generated in Eq. (3). Therefore, from the causal view, in ACL [29], maximizing both the conditional probability of the proxy label given the natural data (i.e., $p(y^R|x)$) and that given adversarial data (i.e., $p(y^R|\tilde{x})$) is equivalent to minimizing the sum of standard and adversarial contrastive losses. $\qquad\square$

## B.2 Proof of Lemma 2

**Lemma 2 (restated).** *AIR in Eq. (7) can be decomposed into two terms as follows:*
$$\mathcal{L}_{\text{AIR}}(B; \theta, \epsilon) = \mathbb{E}_{x \sim p^{do(\tau_i)}(\tilde{x}|x)}[\text{KL}(p^{do(\tau_i)}(y^R|\tilde{x}) \| p^{do(\tau_j)}(y^R|\tilde{x}))]$$
$$+ \mathbb{E}_{x \sim p^{do(\tau_i)}(y^R|\tilde{x})}[\text{KL}(p^{do(\tau_i)}(\tilde{x}|x) \| p^{do(\tau_j)}(\tilde{x}|x))],$$
*where $\mathbb{E}_{x \sim Q^3(x)}[\text{KL}(Q^1(x) \| Q^2(x))]$ is the expectation of KL divergence over $Q^3$ and $Q^1, Q^2, Q^3$ are probability distributions.*

*Proof.* We transform AIR in Eq. (7) as follows:

$$\mathcal{L}_{\text{AIR}}(B;\theta,\epsilon) = \text{KL}\left(p^{do(\tau_i)}(y^R|\tilde{x})p^{do(\tau_i)}(\tilde{x}|x)\|p^{do(\tau_j)}(y^R|\tilde{x})p^{do(\tau_j)}(\tilde{x}|x); B\right) \tag{20}$$

$$= \sum_{x\in B} p^{do(\tau_i)}(y^R|\tilde{x})p^{do(\tau_i)}(\tilde{x}|x) \log \frac{p^{do(\tau_i)}(y^R|\tilde{x})p^{do(\tau_i)}(\tilde{x}|x)}{p^{do(\tau_j)}(y^R|\tilde{x})p^{do(\tau_j)}(\tilde{x}|x)} \tag{21}$$

$$= \sum_{x\in B} p^{do(\tau_i)}(y^R|\tilde{x})p^{do(\tau_i)}(\tilde{x}|x)(\log \frac{p^{do(\tau_i)}(y^R|\tilde{x})}{p^{do(\tau_j)}(y^R|\tilde{x})} + \log \frac{p^{do(\tau_i)}(\tilde{x}|x)}{p^{do(\tau_j)}(\tilde{x}|x)}) \tag{22}$$

$$= \sum_{x\in B} p^{do(\tau_i)}(y^R|\tilde{x}) \log \frac{p^{do(\tau_i)}(y^R|\tilde{x})}{p^{do(\tau_j)}(y^R|\tilde{x})} \cdot p^{do(\tau_i)}(\tilde{x}|x)$$
$$+ \sum_{x\in B} p^{do(\tau_i)}(\tilde{x}|x) \log \frac{p^{do(\tau_i)}(\tilde{x}|x)}{p^{do(\tau_j)}(\tilde{x}|x)} \cdot p^{do(\tau_i)}(y^R|\tilde{x}) \tag{23}$$

$$= \sum_{x\in B} \left(\sum_{x\in\{x\}} p^{do(\tau_i)}(y^R|\tilde{x}) \log \frac{p^{do(\tau_i)}(y^R|\tilde{x})}{p^{do(\tau_j)}(y^R|\tilde{x})}\right) \cdot p^{do(\tau_i)}(\tilde{x}|x)$$
$$+ \sum_{x\in B} \left(\sum_{x\in\{x\}} p^{do(\tau_i)}(\tilde{x}|x) \log \frac{p^{do(\tau_i)}(\tilde{x}|x)}{p^{do(\tau_j)}(\tilde{x}|x)}\right) \cdot p^{do(\tau_i)}(y^R|\tilde{x}) \tag{24}$$

$$= \sum_{x\in B} \text{KL}(p^{do(\tau_i)}(y^R|\tilde{x})\|p^{do(\tau_j)}(y^R|\tilde{x}; \{x\}) \cdot p^{do(\tau_i)}(\tilde{x}|x)$$
$$+ \sum_{x\in B} \text{KL}(p^{do(\tau_i)}(\tilde{x}|x)\|p^{do(\tau_j)}(\tilde{x}|x); \{x\}) \cdot p^{do(\tau_i)}(y^R|\tilde{x}) \tag{25}$$

$$= \mathbb{E}_{x\sim p^{do(\tau_i)}(\tilde{x}|x)}[\text{KL}(p^{do(\tau_i)}(y^R|\tilde{x})\|p^{do(\tau_j)}(y^R|\tilde{x}); \{x\})]$$
$$+ \mathbb{E}_{x\sim p^{do(\tau_i)}(y^R|\tilde{x})}[\text{KL}(p^{do(\tau_i)}(\tilde{x}|x)\|p^{do(\tau_j)}(\tilde{x}|x); \{x\})] \tag{26}$$

$$= \mathbb{E}_{x\sim p^{do(\tau_i)}(\tilde{x}|x)}[\text{KL}(p^{do(\tau_i)}(y^R|\tilde{x})\|p^{do(\tau_j)}(y^R|\tilde{x}))]$$
$$+ \mathbb{E}_{x\sim p^{do(\tau_i)}(y^R|\tilde{x})}[\text{KL}(p^{do(\tau_i)}(\tilde{x}|x)\|p^{do(\tau_j)}(\tilde{x}|x))] \tag{27}$$

Eq. (21) and (25) are obtained by using the KL divergence $\text{KL}(p(x)\|q(x); B) = \sum_{x\in B} p(x) \log \frac{p(x)}{q(x)}$. We obtained Eq. (27) by omiting the term $\{x\}$ in Eq. (26) for simplicity and finished proving Lemma 2. $\square$

## B.3 Proof of Proposition 3

**Proposition 3 (restated).** *AIR implicitly enforces the robust representation to satisfy the following two proxy criteria:*

$$(1) \quad p^{do(\tau_i)}(y^R|\tilde{x}) = p^{do(\tau_j)}(y^R|\tilde{x}), \quad (2) \quad p^{do(\tau_i)}(\tilde{x}|x) = p^{do(\tau_j)}(\tilde{x}|x).$$

*Proof.* To enforce the robust representation to satisfy the two proxy criteria shown in Eq. (28), we need to regulate the ACL with the following regularization

$$\text{KL}\left(p^{do(\tau_i)}(y^R|\tilde{x})\|p^{do(\tau_j)}(y^R|\tilde{x}); B\right) + \text{KL}\left(p^{do(\tau_i)}(\tilde{x}|x)\|p^{do(\tau_j)}(\tilde{x}|x); B\right) \tag{28}$$

$$= \sum_{x\in B} p^{do(\tau_i)}(y^R|\tilde{x}) \log \frac{p^{do(\tau_i)}(y^R|\tilde{x})}{p^{do(\tau_j)}(y^R|\tilde{x})} + \sum_{x\in B} p^{do(\tau_i)}(\tilde{x}|x) \log \frac{p^{do(\tau_i)}(\tilde{x}|x)}{p^{do(\tau_j)}(\tilde{x}|x)} \tag{29}$$

$$= \sum_{x\in B} \text{KL}(p^{do(\tau_i)}(y^R|\tilde{x})\|p^{do(\tau_j)}(y^R|\tilde{x}; \{x\}) + \sum_{x\in B} \text{KL}(p^{do(\tau_i)}(\tilde{x}|x)\|p^{do(\tau_j)}(\tilde{x}|x); \{x\}). \tag{30}$$

Compared with Eq. (25) in the proof of Lemma 2, we can find that AIR also penalizes the same two KL divergence terms while AIR has two extra calibration terms $p^{do(\tau_i)}(\tilde{x}|x)$ and $p^{do(\tau_i)}(y^R|\tilde{x})$ that adjust the confidence of each KL divergence term, respectively. We empirically studied the impact of

the calibration term in Appendix C.8 and found that the calibration term is beneficial to improving the performance of ACL. Therefore, it indicates that AIR implicitly enforces the robust representation to satisfy the two proxy criteria shown in Eq. (28). □

### B.4 Proof of Proposition 4

**Proposition 4 (restated).** *Let* $\mathcal{Y} = \{y_t\}_{t=1}^T$ *be a label set of a downstream classification task,* $\mathcal{Y}^R$ *be a refinement of* $\mathcal{Y}$, *and* $\tilde{x}_t$ *be the adversarial data generated on the downstream task. Assuming that* $\tilde{x}_t \in \mathcal{B}_\epsilon[x]$ *and* $\tilde{x} \in \mathcal{B}_\epsilon[x]$, *we have the following results:*

$$p^{do(\tau_i)}(y^R|\tilde{x}) = p^{do(\tau_j)}(y^R|\tilde{x}) \implies p^{do(\tau_i)}(y_t|\tilde{x}_t) = p^{do(\tau_j)}(y_t|\tilde{x}_t) \quad \forall \tau_i, \tau_j \in \mathcal{T},$$

$$p^{do(\tau_i)}(\tilde{x}|x) = p^{do(\tau_j)}(\tilde{x}|x) \implies p^{do(\tau_i)}(\tilde{x}_t|x) = p^{do(\tau_j)}(\tilde{x}_t|x) \quad \forall \tau_i, \tau_j \in \mathcal{T}.$$

*Proof.* The proof is inspired by Mitrovic et al. [38].

For $t \in \{1, \dots, T\}$, we have

$$p^{do(\tau_i)}(y_t|\tilde{x}_t) = \int p^{do(\tau_i)}(y_t|y^R) p^{do(\tau_i)}(y^R|\tilde{x}_t) dy^R \tag{31}$$

$$= \int p(y_t|y^R) p^{do(\tau_i)}(y^R|\tilde{x}_t) dy^R \tag{32}$$

$$= \int p(y_t|y^R) p^{do(\tau_j)}(y^R|\tilde{x}_t) dy^R \tag{33}$$

$$= p^{do(\tau_j)}(y_t|\tilde{x}_t). \tag{34}$$

Eq. (31) holds due to the fact that $y^R$ is a refinement of $y_t$ according to the definition of the refinement shown in Mitrovic et al. [38]. Eq. (32) holds since the relationship between $y_t$ and $y^R$ is independent of the style, i.e., $p^{do(\tau_i)}(y_t|y^R) = p^{do(\tau_j)}(y_t|y^R)$. Eq. (33) holds because of the assumption that the prediction of adversarial data is independent of the style. Due to that the condition $p^{do(\tau_i)}(y^R|\tilde{x}) = p^{do(\tau_j)}(y^R|\tilde{x})$ holds for any $\tilde{x} \in \mathcal{B}_\epsilon[x]$, thereby, this condition also holds for $\tilde{x}_t \in \mathcal{B}_\epsilon[x]$. Lastly, we obtain that the prediction of adversarial data will be still independent of the style factors on downstream tasks.

Next, due to that the condition $p^{do(\tau_i)}(\tilde{x}|x) = p^{do(\tau_j)}(\tilde{x}|x)$ holds for any $\tilde{x} \in \mathcal{B}_\epsilon[x]$, this condition will hold for $\tilde{x}_t \in \mathcal{B}_\epsilon[x]$ as well, i.e., $p^{do(\tau_i)}(\tilde{x}_t|x) = p^{do(\tau_j)}(\tilde{x}_t|x)$. Therefore, the consistency between adversarial and natural data will be still invariant of the style factors on the downstream tasks. □

## C  Extensive Experimental Details and Results

**Experimental environments.**    We conducted all experiments on Python 3.8.8 (PyTorch 1.13) with NVIDIA RTX A5000 GPUs (CUDA 11.6).

**Pre-training details of ACL [29].**    Following Jiang et al. [29], we leveraged ResNet-18 [25] with the dual batch normalization (BN) [46] as the representation extractor, where one BN is used for the standard branch of the feature extractor and the other BN is used for the adversarial branch, during conducting ACL [29] and its variant DynACL [36]. We pre-trained ResNet-18 models using SGD for 1000 epochs with an initial learning rate 5.0 and a cosine annealing schedule [35]. During pre-training, we set the adversarial budget $\epsilon$ as $8/255$, the hyperparameter $\omega$ as 0.0, and the strength of data augmentation as 1.0. As for the reproduction of the baseline, we used the pre-trained weights published in the GitHub of ACL[45] as the pre-trained representation extractor for finetuning.

**Pre-training details of DynACL [36].**    The training configurations of DynACL [36] followed ACL [29], except for the strength of data augmentation and the hyperparameter $\omega$. We denote the

---

[4]Link of pre-trained weights via ACL on CIFAR-10.
[5]Link of pre-trained weights via ACL on CIFAR-100.

Table 9: The robust/standard test accuracy (%) achieved by ACL-AIR with different $\lambda_1$ and $\lambda_2$.

| $\lambda_1$ \\ $\lambda_2$ | 0.00 | 0.25 | 0.50 | 1.00 |
|---|---|---|---|---|
| 0.00 | 37.39/78.27 | 38.61/79.48 | 38.70/79.96 | 38.76/79.81 |
| 0.25 | 37.55/78.53 | 38.70/79.53 | 38.79/79.65 | 38.73/79.76 |
| 0.50 | 37.51/78.97 | 38.63/79.72 | **38.89/80.03** | 38.82/80.08 |
| 1.00 | 37.04/79.17 | 38.24/79.64 | 38.39/80.09 | 38.77/79.83 |

Table 10: Performance in semi-supervised settings evaluated on the CIFAR-10 task.

| Label ratio | ACL [29] | | ACL-AIR | | DynACL [36] | | DynACL-AIR | |
|---|---|---|---|---|---|---|---|---|
| | AA (%) | SA (%) | AA (%) | SA (%) | AA (%) | SA (%) | AA (%) | SA (%) |
| 1% | 45.63 | 74.84 | **45.97** | **76.63** | 45.78 | 76.89 | **46.25** | **78.57** |
| 10% | 45.68 | 76.30 | **46.17** | **77.52** | 47.08 | 78.22 | **47.41** | **79.94** |

strength of data augmentation and the hyperparameter at epoch $e$ as $\mu_e$ and $\omega_e$ respectively, where

$$\mu_e = 1 - \lfloor \frac{e}{K} \rfloor \cdot \frac{K}{E}, \quad e \in \{0, 1, \ldots, E - 1\} \tag{35}$$

$$\omega_e = \nu \cdot (1 - \mu_e), \quad e \in \{0, 1, \ldots, E - 1\} \tag{36}$$

in which the decay period $K = 50$, the reweighting rate $\nu = 2/3$, the total training epoch $E = 1000$. In our implementation of DynACL, we only take the dynamic strategy and do not take the trick of the stop gradient operation and the momentum encode [24, 11]. As for the reproduction of the baseline, we downloaded the pre-trained weights published in the GitHub of DynACL[678] as the pre-trained representation extractor for finetuning.

**Details of finetuning procedures.** As for SLF and ALF, we fixed the parameters of the representation extractor and only finetuned the linear classifier using the natural training data and adversarial training data respectively. As for AFF, we finetuned all the parameters using the adversarial training data. For all finetuning procedures, we used SGD for 25 epochs for linear finetuning and full finetuning respectively and momentum is set as $2e - 4$. The initial learning of linear finetuning is set as $0.01$ on CIAFR-10, $0.05$ on CIFAR-100, and $0.1$ on STL-10. The adversarial budget is fixed as $8/255$ for ALF and AFF. In practice, we used the finetuning code published in the GitHub of DynACL for implementing finetuning procedures.

### C.1 Performance in Semi-Supervised Settings [1]

Table 10 reports the performance evaluated in semi-supervised settings [1] during the finetuning procedure. In semi-supervised settings, following ACL [29] and DynACL [36], we first standardly finetuned the pre-trained model using the labelled data and then generated the pseudo labels using the standardly finetuned model. Then, we finetuned the model using the data with pseudo labels as well as the labelled data via AFF. The results validate that AIR ($\lambda_1 = 0.5, \lambda_2 = 0.5$) can consistently enhance both robust and standard test accuracy of ACL and its variant DynACL in semi-supervised settings.

### C.2 Impact of the Hyper-Parameters $\lambda_1$ and $\lambda_2$

We show the performance achieved by ACL-AIR of different $\lambda_1 \in \{0.00, 0.25, 0.50, 1.00\}$ and $\lambda_2 \in \{0.00, 0.25, 0.50, 1.00\}$ on the CIFAR-10 task in Table 9. We can notice that when only leveraging adversarial regularization ($\lambda_1 = 0, \lambda_2 > 0$), both standard and robust test accuracy gain significant improvement, which indicates that adversarial regularization serves an important role in improving the standard generalization and robustness of ACL. By incorporating adversarial and standard regularization together (when both $\lambda_1 > 0$ and $\lambda_2 > 0$), the standard generalization of ACL gets further improved. Table 9 guides us to set $\lambda_1 = 0.5$ and $\lambda_2 = 0.5$ since it can yield the best performance.

### C.3 Incorporating AIR with Two Extra Variants of ACL

Here, we show extra results to validate that our proposed AIR can enhance two extra variants of ACL, including AdvCL [22] and AdvCL with A-InfoNCE [50].

---

[6]Link of pre-trained weights via DynACL on CIFAR-10.

[7]Link of pre-trained weights via DynACL on CIFAR-100.

[8]Link of pre-trained weights via DynACL on STL-10.

Table 11: Self-task adversarial robustness transferability evaluated on CIFAR-10 using various backbone networks including ResNet-34, ResNet-50, and WRN-28-10.

| Network | Pre-training | AA (%) | SA (%) |
|---|---|---|---|
| ResNet-34 | DynACL [36] | 47.01 | 78.84 |
| | DynACL-AIR | **47.56** | **80.67** |
| ResNet-50 | DynACL [36] | 47.19 | 79.65 |
| | DynACL-AIR | **47.82** | **81.27** |
| WRN-28-10 | DynACL [36] | 41.13 | 71.23 |
| | DynACL-AIR | **43.86** | **73.41** |

Table 12: Self-task adversarial robustness transferability evaluated on CIFAR-10 achieved by AdvCL [22], AdvCL-AIR, A-InfoNCE [50], and A-InfoNCE-AIR.

| Pre-training | AdvCL [22] | AdvCL-AIR | A-InfoNCE [50] | A-InfoNCE-AIR |
|---|---|---|---|---|
| AA (%) | 42.58 | **43.24** | 42.68 | **42.84** |
| SA (%) | 80.78 | **81.53** | 83.18 | **83.99** |

Table 13: Self-task adversarial robustness transferability evaluated on the CIFAR-10 dataset via SLF under various attacks including APGD-CE [13], APGD-DLR [13], FAB [14], and Square Attack [2].

| Pre-training | APGD-CE (%) | APGD-DLR (%) | FAB (%) | Square Attack (%) |
|---|---|---|---|---|
| ACL [29] | 39.66 | 41.18 | 39.27 | 48.98 |
| ACL-AIR | **40.39** | **40.98** | **41.01** | **49.80** |
| DynACL [36] | 46.28 | 46.41 | 45.56 | 50.06 |
| DynACL-AIR | **47.05** | **47.29** | **46.02** | **50.58** |

AdvCL [22] leverages an extra contrastive view of high-frequency components and the pseudo labels generated by the clustering method. We leveraged our proposed method IR ($\lambda_1 = 0.5, \lambda_2 = 0.5$) to further improve the performance of AdvCL. Yu et al. [50] proposed an asymmetric InfoNCE objective (A-InfoNCE) that treats adversaries as inferior positives that induce weaker learning signals, or as hard negatives exhibiting higher contrast to other negative samples.

We used the code provide by Yu et al. [50] in their GitHub[9] to implement the pre-training of AdvCL as well as A-InfoNCE, and the finetuning procedure. In Table 12, we show that our proposed AIR ($\lambda_1 = 0.5, \lambda_2 = 0.5$) can consistently improve the robust and standard test accuracy of two extra variants [22, 50].

### C.4 Applicability with Different Backbone Networks

In this subsection, we demonstrate that DynACL-AIR can consistently improve the performance on ResNet-34, ResNet-50 [25], and WRN-28-10 [51]. We followed the same training configurations of pre-training and finetuning (SLF) in Section 4 except for the backbone network. Table 11 validates that IR can further enhance the robust and standard test accuracy of DynACL on various backbone networks.

We observe that WRN-28-10 yields a worse performance compared to ResNet-50. We conjecture that it is because we set the batch size to 128 during pre-training due to the limitation of our GPU memory. We believe using a larger batch size during pre-training can further improve the performance of WRN-28-10 according to Chen et al. [9].

### C.5 Robustness Evaluation under Various Attacks

In this subsection, we conducted the robustness self-transferability against three strong white-box attacks (APGD-CE [13], APGD-DLR [13] and FAB [14]) and one strong black-box attack (i.e., Square Attack [2]). We evaluate the robustness on the CIFAR-10 dataset via SLF and report the results in Table 13. The results validate that our proposed method can consistently improve robust test accuracy over various adversaries.

### C.6 Test Accuracy under Each Type of Common Corruption

We report the test accuracy under each type of common corruption [26] with corruption severity being fixed as 5 in Table 14. We used pre-trained models on CIFAR-10 via DynACL and DynACL-AIR.

---

[9]GitHub provided by Yu et al. [50].

Table 14: Test accuracy evaluated on CIFAR-10-C under each type of common corruptions with corruption severity being fixed as 5 of CIFAR-10 pre-trained models after SLF and AFF.

| Pre-training | Finetuning | Noise | | | Blur | | | | Weather | | | | Digital | | | |
|---|---|---|---|---|---|---|---|---|---|---|---|---|---|---|---|---|
| | | Gaussian | Shot | Impulse | Defocus | Glass | Motion | Zoom | Snow | Frost | Fog | Bright | Contrast | Elastic | Pixel | JPEG |
| DynACL [36] | SLF | 68.14 | 68.22 | 62.76 | 69.73 | 68.84 | 67.01 | 71.86 | 64.73 | 62.36 | 26.24 | 66.53 | 17.66 | 70.64 | 72.44 | 73.63 |
| DynACL-AIR | SLF | 70.10 | 70.47 | 64.40 | 69.79 | 71.18 | 67.90 | 72.10 | 66.72 | 63.94 | 30.16 | 67.27 | 18.21 | 72.43 | 75.17 | 76.80 |
| DynACL [36] | AFF | 74.02 | 74.62 | 66.87 | 71.44 | 73.21 | 69.22 | 73.68 | 67.83 | 66.29 | 27.59 | 69.27 | 18.72 | 74.60 | 77.50 | 79.05 |
| DynACL-AIR | AFF | 74.54 | 75.13 | 67.06 | 72.63 | 74.78 | 70.39 | 75.66 | 69.07 | 67.47 | 28.74 | 71.24 | 19.02 | 75.85 | 79.09 | 80.64 |

Table 15: Standard deviation of robustness self-transferability against common corruptions reported in Table 2.

| Dataset | Pre-training | SLF | | | ALF | | | AFF | | |
|---|---|---|---|---|---|---|---|---|---|---|
| | | CS-1 | CS-3 | CS-5 | CS-1 | CS-3 | CS-5 | CS-1 | CS-3 | CS-5 |
| CIFAR-10 | ACL [29] | 0.06 | 0.25 | 0.13 | 0.13 | 0.17 | 0.10 | 0.11 | 0.17 | 0.21 |
| | ACL-AIR | 0.03 | 0.02 | 0.02 | 0.07 | 0.09 | 0.08 | 0.16 | 0.20 | 0.14 |
| | DynACL [36] | 0.02 | 0.01 | 0.02 | 0.07 | 0.16 | 0.09 | 0.13 | 0.20 | 0.18 |
| | DynACL-AIR | 0.01 | 0.03 | 0.03 | 0.07 | 0.11 | 0.14 | 0.16 | 0.11 | 0.10 |
| CIFAR-100 | ACL [29] | 0.01 | 0.09 | 0.02 | 0.12 | 0.14 | 0.11 | 0.09 | 0.04 | 0.17 |
| | ACL-AIR | 0.04 | 0.02 | 0.02 | 0.13 | 0.09 | 0.04 | 0.17 | 0.04 | 0.16 |
| | DynACL [36] | 0.03 | 0.02 | 0.01 | 0.14 | 0.07 | 0.12 | 0.08 | 0.09 | 0.16 |
| | DynACL-AIR | 0.08 | 0.05 | 0.02 | 0.08 | 0.12 | 0.05 | 0.16 | 0.16 | 0.10 |

Table 16: Standard deviation of robustness self-transferability against common corruptions reported in Table 4.

| $\mathcal{D}_1 \to \mathcal{D}_2$ | Pre-training | SLF | | | ALF | | | AFF | | |
|---|---|---|---|---|---|---|---|---|---|---|
| | | CS-1 | CS-3 | CS-5 | CS-1 | CS-3 | CS-5 | CS-1 | CS-3 | CS-5 |
| CIFAR-10 $\to$ CIFAR-100 | ACL [29] | 0.05 | 0.02 | 0.03 | 0.07 | 0.09 | 0.05 | 0.15 | 0.10 | 0.31 |
| | ACL-AIR | 0.01 | 0.04 | 0.04 | 0.09 | 0.12 | 0.10 | 0.11 | 0.16 | 0.15 |
| | DynACL [36] | 0.04 | 0.04 | 0.05 | 0.11 | 0.07 | 0.07 | 0.19 | 0.18 | 0.10 |
| | DynACL-AIR | 0.03 | 0.02 | 0.03 | 0.12 | 0.13 | 0.09 | 0.18 | 0.04 | 0.07 |
| CIFAR-100 $\to$ CIFAR-10 | ACL [29] | 0.03 | 0.03 | 0.03 | 0.08 | 0.09 | 0.14 | 0.08 | 0.07 | 0.15 |
| | ACL-AIR | 0.03 | 0.02 | 0.03 | 0.09 | 0.15 | 0.13 | 0.13 | 0.14 | 0.20 |
| | DynACL [36] | 0.03 | 0.01 | 0.02 | 0.11 | 0.12 | 0.14 | 0.05 | 0.14 | 0.19 |
| | DynACL-AIR | 0.02 | 0.08 | 0.02 | 0.06 | 0.08 | 0.09 | 0.14 | 0.21 | 0.17 |

Table 17: Impact of calibration terms evaluated on the CIFAR-10 task.

| Pre-training | Calibration | SLF | | ALF | | AFF | |
|---|---|---|---|---|---|---|---|
| | | AA (%) | SA (%) | AA (%) | SA (%) | AA (%) | SA (%) |
| ACL-AIR | × | 38.55 | 79.80 | 40.80 | 77.57 | 49.51 | 81.95 |
| ACL-AIR | √ | **38.70** | **79.96** | **41.09** | **77.99** | **49.59** | **82.30** |
| DynACL-AIR | × | 45.09 | 77.79 | 46.01 | 76.12 | 50.54 | 82.35 |
| DynACL-AIR | √ | **45.23** | **78.01** | **46.12** | **77.01** | **50.66** | **82.62** |

Experimental settings are the same as Section 4. The results validate that IR can consistently improve the test accuracy under each type of common corruption.

## C.7 Standard Deviation of Test Accuracy under Common Corruptions

Here, we provide the standard deviation of the test accuracy under common corruptions in Tables 15 and 16.

## C.8 Impact of Calibration Terms

In the proof of Proposition 3, we show that there exist calibration terms $p^{do(\tau_i)}(\tilde{x}|x)$ and $p^{do(\tau_i)}(y^R|\tilde{x})$ in AIR. Here, we empirically investigate the impact of calibration on the performance in Table 17. The results empirically validate that the calibration term is an essential component of AIR to adjust the confidence of the KL divergence and further enhance the performance of ACL.

# D Limitations

One of the limitations is that similar to previous works [29, 22, 50, 36], our proposed AIR is not helpful in improving the efficiency and scalability of ACL methods. Therefore, robust pre-training via

ACL-AIR is still unfriendly to the environment due to emitting much carbon dioxide and consuming much electricity. Besides, applying ACL-AIR to large-scale datasets such as ImageNet-21K [41] is still computationally prohibitive with limited GPU sources.

## E    Possible Negative Societal Impacts

Our proposed method aims to improve the performance of robust self-supervised pre-training methods [29, 22, 50, 36]. The robust self-supervised pre-training procedure is extremely time-consuming since it needs to spend much time generating adversarial training data, which thus leads to consuming a lot of electricity and emitting lots of carbon dioxide. Unfortunately, our proposed method is not helpful in improving the efficiency of ACL. Consequently, our study could exacerbate the greenhouse effect and be not conducive to environmental protection.

