# OpenReview forum: "Enhancing Adversarial Contrastive Learning via Adversarial Invariant Regularization"
_NeurIPS.cc/2023/Conference — NeurIPS 2023 poster_

### Official Review · Reviewer_Ai1a · 2023-06-19

**Soundness:** 3 good
**Presentation:** 3 good
**Contribution:** 2 fair
**Rating:** 8
**Confidence:** 4

**Summary:**

The authors propose AIR to regulate the the process of contrastive learning. They first analyze the causal graph of contrastive learning under the adversarial setting, and try to enforce  $p\left(y^R | \tilde x\right)\cdot p\left(\tilde x | x\right)$ (Eq.4) to be invariant under different interventions. Intuitively, this regularizer would force the crafted adversarial examples to be style-independent and thus the learned representations shall enjoy better robustness.

**Strengths:**

The core idea of this paper is technically novel and clearly presented.

Experiments are reported by applying the proposed AIR / SIR terms to ACL / DynACL, and consistent performance gains are observed. Ablation studies for different values of $\lambda_1$ and $\lambda_2$ are also provided, to verify the effectiveness of the AIR term itself.

The authors also provide theoretical results to justify the rationality of the AIR term.

**Weaknesses:**

The proposed method would require more tunning since it incorporates two more hyper-parameters $\lambda_1$ and $\lambda_2$, and we can see from Table 7 that these two hyper-parameters interact in a non-trival way. For example, when compared with the case that only AIR is leveraged ($\lambda_1=0,\ \lambda_2>0$), incorporating AIR with SIR (i.e., $\lambda_1>0,\ \lambda_2>0$) could result in better **robust accuracy**.

Little type in Eq. 8: two augmentations in KL divergence should be different.

--- Update ---
After reading additional experimented provieded by the authors, I decide to raise my score to 8.

**Questions:**

See "Weaknesses" part.

**Limitations:**

N.A.

---

> ### Author Rebuttal · Authors · 2023-08-04
>
> Many thanks for your supportive comments! Please find our responses below.
>
> > 1. [Reply to W1] Our proposed AIR is less sensitive to the hyper-parameter (HP) than SIR.
>  Automatic schedule of the HP could be a good extension to our work.
>
> We provide an ablation study of HP in Table 7. In ***Point 4*** of Rebuttal Highlights, we provide a detailed analysis to show that AIR is less sensitive to HP than SIR. We would like to treat finding an automatic scheduler of HP as future work.
>
> > 2. [Comment on typo] Thanks for pointing out this typo. We have corrected this typo in the revision.

---

### Official Review · Reviewer_paeS · 2023-06-22

**Soundness:** 3 good
**Presentation:** 3 good
**Contribution:** 3 good
**Rating:** 6
**Confidence:** 4

**Summary:**

This paper proposes a novel adversarial contrastive learning method, which introduces causal reasoning method to obtain robust feature representation and enforce independence from style factors. The idea is simple and effective. In addition, the experiments are sufficient to prove the effectiveness of the proposed method. The source codes are released.

**Strengths:**

1. The idea is interesting and easy to understand.
2. The theoretical analysis is detailed and proves the effectiveness of the proposed method.


**Weaknesses:**

1. The improvement of the proposed method is limited.
2. Without causal reasoning, this paper only proposes two distribution alignment regularizations in the adversarial samples to obtain robust feature representation. Thus, this paper is an incremental work based on ACL and DynACL.
3. I want to see the performance of this paper in the large-scale datasets, such as sub-imagenet.

**Questions:**

Shown as Weaknesses.

**Limitations:**

Shown as Weaknesses.

---

> ### Author Rebuttal · Authors · 2023-08-04
>
> Many thanks for your constructive comments! Please find our replies below.
>
> > 1. [Reply to W1] In ***Table A*** of Rebuttal Highlights, we report the p-value obtained by conducting a Student's t-test to show that our improvement is **significant**.
>
> ***Table A*** shows that the p-value is consistently much smaller than 0.05, which validates that the performance gain of our proposed method is significant.
>
> > 2. [Reply to W2] We argue it is **non-trivial** to derive our proposed regularization term AIR. We use the causal graph of ACL to provide a unified framework.  Kindly note that our proposed regularization is a handy plug-in that can improve all existing ACL methods.
>
> Without causal reasoning, we cannot build the causal graph of ACL. Further, without ACL's causal graph, we cannot obtain Eq. (4) according to the unique path $x \rightarrow \tilde{x} \rightarrow y^R$ shown in ACL's causal graph. Besides, without causal reasoning of ACL's causal graph, we cannot propose the criterion shown in Eq. (5) and further propose AIR. Therefore, it is non-trivial to derive our proposed regularization term AIR.
>
> Our proposed method can complement all ACL-style pre-training methods such as ACL, DynACL, AdvCL [1], and A-InfoNCE [2], which is validated in the main paper and Appendix C.5. Therefore, our proposed method is a useful plug-in for all ACL-style pre-training methods.
>
> > 3. [Reply to W3] We provide the results of conducting DynACL with IR on **Imagenette datasets** which is a subset of ImageNet-1K.
>
> To our knowledge, there is no reported result of ACL on large-scale datasets in existing papers. We report new results of the performance evaluated on Imagenette in ***Table B.1*** and ***Table B.2*** of Rebuttal Highlights. The results validate that our method is effective in sub-imagenet.
>
> *References*
>
> [1] When does contrastive learning preserve adversarial robustness from pretraining to finetuning? Fan et al., NeurIPS 2021.\
> [2] Adversarial contrastive learning via asymmetric infonce. Yu et al., ECCV 2022.

---

> ### Author Response · Authors · 2023-08-15
> **Would our responses properly solve your concerns?**
>
> Dear Reviewer **paeS**
>
> As "Author-Reviewer Discussions are going to end on 21st, August, would we know whether our responses satisfactorily address your concerns?
>
> Best wishes,\
> Authors

---

> > ### Comment · Reviewer_paeS · 2023-08-17
> >
> > Dear authors,
> >
> > Many thanks for your responses.
> >
> > After reading your responses and reviews from other reviewers, my concerns are addressed well. Thus, I will support this paper for acceptance.
> >
> > Best regards,
> >
> > Reviewer paeS

---

### Official Review · Reviewer_ZaSM · 2023-07-04

**Soundness:** 3 good
**Presentation:** 3 good
**Contribution:** 3 good
**Rating:** 6
**Confidence:** 4

**Summary:**

This paper proposes to tackle adversarial contrastive learning via causal reasoning. Specifically, the authors discuss the scenario where adversarial examples are involved in standard invariant regularization. The authors also provide analysis of proposed invariant regularization to justify the rationality. The experiments on different datasets show improvement compared with baselines.

**Strengths:**

1. The paper is well-written and easy to follow.
2. Taking adversarial example in the SIR via Markov condition seems natural and the analysis could provide some insights.
3. The experiments on various datasets show improvement.


**Weaknesses:**

I have several concerns:

1. The comparison with other baselines is missing in Table 4, such as ACL with SIR and DynACL with SIR.
2. The authors conduct experiments on ResNet family. However, WideResNet backbone networks are always evaluated in the field of adversarial robustness. I wonder whether the superiority of IR can be generalized to different networks.

Minors:

It seems a typo in Eq. 8, the KL divergence should be computed between $\tau_i$ and $\tau_j$.



**Questions:**

1. Please include the comparison with baseline in Table 4.
2. Please include more evaluation on WideResNet.


**Limitations:**

The authors have discussed the limitations.

---

> ### Author Rebuttal · Authors · 2023-08-04
>
> Many thanks for your positive comments! Please find our replies below.
>
> > 1. [Reply to Q1] We demonstrate the results of the baseline as follows. We will update them in revision.
>
> The results show that our proposed IR consistently performs better than the baseline.
>
> |Label ratio = 1\%  | ACL | ACL with SIR| ACL with IR |
> |---|---|---|---|
> |AA (\%) | 45.63 | 45.71 | 45.97 |
> |SA (\%) | 74.84 | 74.99 | 76.63 |
>
> |Label ratio = 10\%  | ACL | ACL with SIR| ACL with IR |
> |---|---|---|---|
> |AA (\%) | 45.68 | 45.79 | 46.17 |
> |SA (\%) | 76.30 | 76.67 | 77.52 |
>
> |Label ratio = 1\%  | DynACL | DynACL with SIR| DynACL with IR |
> |---|---|---|---|
> |AA (\%) | 45.78 | 45.93 | 46.25 |
> |SA (\%) | 76.89 | 77.34 | 78.57 |
>
> |Label ratio = 10\%  | DynACL | DynACL with SIR| DynACL with IR |
> |---|---|---|---|
> |AA (\%) | 47.08 | 47.12 | 47.41 |
> |SA (\%) | 78.22 | 78.84 | 79.94 |
>
>
> > 2. [Reply to Q2] We provide the results evaluated on the Wide ResNet with a depth 28 and width 10 (WRN-28-10) as follows.
>
> We pre-trained WRN-28-10 via DynACL and DynACL with IR on CIFAR-10 and then evaluated the performance on CIFAR-10 via SLF. The results are shown in the following table. It validates that our method is effective in various models.
>
> | WRN-28-10 | SLF on CIFAR-10 | SLF on CIFAR-10 |
> |---|---|---|
> | Methods | AA (\%) | SA (\%)
> | DynACL | 41.13 | 71.23 |
> |DynACL with IR| 43.86 | 73.41 |
>
> Note that we set the batch size to 128 during pre-training due to the limitation of our GPU memory. We believe using a larger batch size during pre-training can further improve the performance of WRN-28-10 since [1].
>
> > 3. [Comment on typo] Thanks for pointing out this typo. We have corrected this typo in the revision.
>
>
> *References*
>
> [1] Chen, Ting, et al. "A simple framework for contrastive learning of visual representations." International conference on machine learning. PMLR, 2020.

---

### Official Review · Reviewer_fVht · 2023-07-07

**Soundness:** 2 fair
**Presentation:** 3 good
**Contribution:** 2 fair
**Rating:** 4
**Confidence:** 4

**Summary:**

This paper proposed a method to advance adversarial contrastive learning by utilizing a technique called causal reasoning. The adversarial invariant regularization (AIR) proposed in this paper demonstrated a style factor. Additionally, the effectiveness of the proposed method was empirically shown using CIFAR10 and CIFAR100.

**Strengths:**

- Theoretically, it explained how the KL divergence can provide invariant regularization.
- The paper is well organized and most explanations are clearly written and easy to understand the approach and the theoretical explanations.

**Weaknesses:**

- The theoretical analysis of this paper (section 3.2 and section 3.3) appears to apply adversarial examples to the theorem shown in paper [1], which seems closer to an application rather than a novel combination, resulting in a perceived lack of originality.
- Although most theorems and reasoning methods are well explained, expressing which input types the KL divergence loss of SIR and AIR applies to would be more effective for reproducing this paper through implementation.
- Furthermore, the performance gain shown in robustness experiments from the proposed regularization is considered to have low significance as it is less than 1% in most datasets and tasks.
- Also, there is insufficient explanation as to why the proposed method is more helpful for self-supervised adversarial robustness than existing methods, making it difficult to verify the experimental significance of the proposed method. Especially in the experimental results, AIR helps both clean performance and robustness compared to SIR, but there is no clear justification for why these two components should be used simultaneously.
- It is unclear why a model trained with SIR and AIR regularization for adversarial x and natural x performs well against common corruption.
- As it's a self-supervised pretraining method, the fact that experiments were only conducted on small datasets (CIFAR10, CIFAR100) leaves concerns about its applicability.

[1] Mitrovic et al., REPRESENTATION LEARNING VIA INVARIANT CAUSAL MECHANISMS, ICLR 2021

**Questions:**

- It would be good to mention y^R in the main text.
- In equation 3, adversarial for x_k^i and x_k^j is expressed in one formula. I'm curious if x_k^j also gets its perturbation updated when creating the adversarial for x_k^i.
- It would be great to mention how much performance difference there is, how much slower the speed is, and how much the transferability improves when compared to Supervised Adversarial Training (Madry, TRADES).

**Limitations:**

- There is a limitation in the hyperparameter tuning of lambda1 and lambda2.
- There is a limitation that the gain in performance is marginal.

---

> ### Author Rebuttal · Authors · 2023-08-04
>
> Many thanks for your constructive comments! Please find our replies below.
>
> > 1. [Reply to W1] We argue that our theoretical analysis is **non-trivial**.
>
> Directly applying adversarial data to paper [1] cannot obtain AIR.  It is because SIR [1] in Eq. (8) aims to enforce $p(y^R|x)$ to be style-independent; however, AIR in Eq. (7) aims to enforce $p(y^R|\tilde{x})p(\tilde{x}|x)$, instead of simply applying adversarial data into SIR (i.e., $p(y^R|\tilde{x})$), to be style-independent.
>
> Besides, our theoretical analysis in Section 2.3 is unique to AIR. We explain AIR in Proposition 3 based on AIR's decomposition shown in Lemma 2, as well as analyze the generalization ability of the style-independence property of robust representations via AIR in Proposition 4.
>
> > 2. [Reply to W2] We explain inputs below and provide source code in Anonymized GitHub (link is in Abstract).
>
> AIR uses $p(y^R|\tilde{x})p(\tilde{x}|x)$ while SIR uses $p(y^R|x)$ under two different augmentations as the inputs. $p(y^R|\tilde{x})$ or $p(\tilde{x}|x)$ in Eq. (6) is calculated as a normalized representational distance (RD) between an original or augmented view of natural data and their adversarial variants by softmax function (SM). $p(y^R|x)$ in Eq. (8) is normalized RD between the original and augmented view of natural data by SM.
>
> > 3. [Reply to W3] In ***Table A*** of Rebuttal Highlights, we report the p-value obtained by conducting a Student's t-test to show that our method gains **significant** improvement.
>
> ***Table A*** shows that the p-value is consistently much smaller than 0.05, which validates that the performance gain of our proposed method is significant.
>
> > 4. [Reply to W4] AIR is a simple plug-in that can help all existing ACL methods. We empirically find that the incorporation of SIR and AIR can obtain SOTA performance.
>
> The embedding should be exempted from nuisance style factors for better transferability [1].
> AIR is a useful plug-in to enhance ACL methods by regulating robust representations, and SIR regulates the standard representations.
> Incorporating AIR and SIR can regulate both standard and robust representations to be style-independent, thus improving both natural generalization and adversarial robustness in downstream tasks.
> Experiments in the main paper and Appendix C.5 validate AIR improved ACL, DynCAL, AdvCL [2], and A-InfoNCE [3] significantly.
>
> > 5. [Reply to W5] AIR and SIR help to find the style-invariant correlations among standard and robust representations across different distributions, which could enhance the robustness against common corruptions.
>
> Proposition 4 in our paper and Theorem 1 in Paper [1] indicate that the style-independent property brought by AIR and SIR is generalizable to downstream tasks. Therefore, it means that AIR and SIR help to find the style-invariant correlations among standard and robust representations across different distributions. It is similar to the objective of invariant risk minimization [4] which can yield substantial improvement in robustness against perturbations incurred by distribution shifts. Therefore, IR could help enhance robustness against common corruption.
>
> > 6. [Reply to W6] We also conducted experiments on **STL10 datasets** in the main paper. Here, we provide the new results by conducting on **Imagenette datasets** which is a subset of ImageNet-1K.
>
> To our knowledge, there is no reported result of ACL on large-scale datasets. We report new results of the performance evaluated on Imagenette in ***Table B.1*** and ***Table B.2*** of Rebuttal Highlights. The results validate that our method is still effective in sub-imagenet.
>
> > 7. [Reply to Q1] We intuitively explain $y^R$ in the main text (refer to Line 144) and a formal formulation of $y^R$ is shown in Eq. (10) in Appendix B.1. In the context of ACL, each augmented data point's proxy label $y^R$ refers to the index of its augmented variant in a minibatch. We will mention the formulation of $y^R$ in the main paper in revisions.
>
> > 8. [Reply to Q2] We formulate each step of the PGD attack following ACL.
>
> Given an initial pair $(x^{i,(0)}, x^{j,(0)})$, PGD step $\kappa \in \mathbb{N}$, step size $\rho > 0$, adversarial budget $\epsilon \geq 0$, PGD iteratively update the pair from $t=0$ to $\kappa-1$ as follows: $x^{u,(t+1)} =  \Pi_{B_{\epsilon}[x^{u,(0)}]} ( x^{u,(t)} +\rho \cdot \mathrm{sign} (\nabla_{x^{u,(t)}} \ell_\mathrm{CL}(x^{i,(t)}, x^{j,(t)})   )$, where $u \in \\{ i,j\\}$ and $\Pi_{B_{\epsilon}[x^{u,(0)}]}$ projects the adversarial data into the epsilon ball around the initial point. At each step $t$, both $x^{i,(t)}$ and $x^{j,(t)}$ are updated.
>
> > 9. [Reply to Q3] We provide a comparison between DynACL with IR and supervised pre-training (SPT) including Madry and TRADES in ***Table C*** of Rebuttal Highlights.
>
> (1) *Speed*. DynACL with IR consumes more pre-training time compared with SPT.
>
> (2) *Performance*. DynACL with IR achieves worse performance on CIFAR-10 than SPT via SLF.
>
> (3) *Transferability*. DynACL with IR achieves better transferability to CIFAR-100 than SPT via SLF.
>
> Due to limited space, we provide a detailed analysis of these phenomena in ***Point 3*** of Rebuttal Highlights.
>
> > 10. [Reply to L1] Our proposed AIR is less sensitive to the hyper-parameter (HP) than SIR. We treat how to automatically schedule the HP as future work.
>
> We provide an ablation study of HP in Table 7. In ***Point 4*** of Rebuttal Highlights, we provide a detailed analysis to show that AIR is less sensitive to HP than SIR. We would like to treat finding an automatic scheduler of HP as future work.
>
> *References*
>
> [1] Representation learning via invariant causal mechanisms. Mitrovic et al., ICLR 2021.\
> [2] When does contrastive learning preserve adversarial robustness from pretraining to finetuning? Fan et al., NeurIPS 2021.\
> [3] Adversarial contrastive learning via asymmetric infonce. Yu et al., ECCV 2022.\
> [4] Invariant risk minimization. Arjovsky et al., 2020.

---

> ### Author Response · Authors · 2023-08-15
> **Would our responses properly solve your concerns?**
>
> Dear Reviewer **fVht**,
>
> As "Author-Reviewer Discussions are going to end on 21st, August, would we know whether our responses satisfactorily address your concerns?
>
> Best wishes,\
> Authors

---

> > ### Comment · Reviewer_fVht · 2023-08-16
> >
> > Thank you to the authors for their detailed response and additional experiments. I appreciate the effort to address the initial concerns. However, after reviewing the updates, I still have several questions regarding the paper:
> >
> > 1. I'm not entirely convinced about the non-trivial nature of the analysis presented. While such perceptions might vary among reviewers, I would appreciate an opportunity to discuss this aspect with fellow reviewers and the ACs. At this juncture, the claim that AIR represents a novel contribution seems a bit overstated.
> >
> > 3. The response lacks specific details about the sample size used for the t-test and the method employed. A small p-value alone isn't sufficiently informative. Is the performance in Table 1 based on a single pretrained checkpoint with multiple re-runs of the linear layer, or are multiple pretrained checkpoints involved? I kindly ask the authors to explain the number of trials used for the t-test and provide further details. Without such clarity, the performance's significance remains ambiguous.
> >
> > 4. Unfortunately, this recent response didn't fully address my initial queries. There's a lack of detailed explanation regarding why the proposed approach is superior to AIR. For instance, as per Tables 5 or 6, using AIR seems to enhance both clarity and robustness more significantly compared to SIR. Why then is there a need to employ both SIR and AIR? Why not solely focus on enhancing AIR?
> >
> > 5. Do the authors imply that the styles of adversarial perturbation and common corruptions are treated similarly in IR? I'm having difficulty agreeing with this perspective. Adversarial perturbation typically introduces noise to the original image 'x,' whereas common corruption may introduce noise or even result in geometric shifts. Given the distinct nature of these perturbations, I'm still uncertain about how IR achieves superior performance in both scenarios.
> >
> > If these concerns remain unresolved after our discussions, I might reconsider and adjust my score to a 3.

---

> > > ### Author Response · Authors · 2023-08-17
> > > **Reply to Reviewer fVht**
> > >
> > > Thanks for your replies. Please find our responses below.
> > >
> > > > [Reply to Q1] Kindly note that, to the best of our knowledge, we are the first effort to connect causal reasoning with ACL. We construct a novel causal graph of ACL and provide the theorem to justify the rationality of our constructed causal graph of ACL.
> > >
> > > > [Reply to Q2] Details of the t-test.
> > >
> > > In Table 1 (a.k.a. ***Table A***), for **baselines** (i.e., ACL and DynACL) and **our proposed method** (i.e., ACL with IR and DynACL with IR), we used a single pre-trained checkpoint and repeated the finetuning procedure three times. Therefore, for each baseline and each dataset, we have **three** results of robust test accuracy and **three** results of standard test accuracy. Note that, for baselines, we used the public pre-trained checkpoints downloaded from their official GitHub.
> > >
> > > To conduct t-tests, we used the following code:
> > > ```
> > > from scipy import stats
> > > a = the list of results of the baseline
> > > b = the list of results of our proposed method
> > > if stats.levene(a, b).pvalue > 0.05: # check whether the variance is the same.
> > >     print(stats.ttest_ind(a,b,equal_var=True).pvalue) # check whether there is significant difference between two distributions.
> > > else:
> > >     print(stats.ttest_ind(a,b,equal_var=False).pvalue)
> > > ```
> > >
> > > On each dataset, we conducted a t-test between the three results of robust/standard test accuracy achieved by the baseline and the three results of robust/standard test accuracy achieved by our proposed method. Then, we reported the corresponding p-value in ***Table A***. We can observe that the p-value is far smaller than 0.05, which means that the t-test rejects the null hypothesis and our results are significantly better than the baseline.
> > >
> > >
> > > > [Reply to Q3] Perhaps there is a misunderstanding. Kindly note that SIR is the method of [1]. **AIR is *our* proposed regularization**. IR means AIR+SIR.
> > >
> > > As you mentioned, our proposed AIR is apparently more effective than SIR since AIR can enhance both standard and robust test accuracy more significantly compared to SIR.
> > >
> > > We empirically found that IR slightly improves performance compared to AIR. Therefore, we choose to use SIR and our proposed AIR together.
> > >
> > > > [Reply to Q4] IR regulates the representations to be invariant of style factors, thus enhancing the robustness in downstream tasks.
> > >
> > > We can treat adversarial attacks and common corruptions as style factors which are functions to manipulate the original data without changing its semantics. Adversarial attacks add imperceptible input perturbations to the original data while maintaining the original semantics. Common corruptions [2] used 15 types of algorithms from noise, blur, weather, and digital categories to generate input perturbations without modifying the semantics of the original data.
> > >
> > > IR aims to regulate the representations to be invariant of style factors. Therefore, IR helps to make the representations robust against style factors that do not change the semantics, thus improving robustness against adversarial robustness and common corruptions.
> > >
> > > *References*\
> > > [1] Representation learning via invariant causal mechanisms. Mitrovic et al., ICLR 2021.\
> > > [2] Benchmarking Neural Network Robustness to Common Corruptions and Perturbations. Hendrycks et al., ICLR 2019.

---

> > > > ### Comment · Area_Chair_waST · 2023-08-19
> > > > **Remaining concerns**
> > > >
> > > > Dear Reviewer fVht,
> > > >
> > > > Are the authors' responses effective in addressing your concerns? Thank you.
> > > >
> > > > AC

---

> > > > ### Comment · Reviewer_fVht · 2023-08-19
> > > >
> > > > Thank you for your detailed responses. Your feedback has provided clarity and addressed some of my initial concerns.
> > > >
> > > > > In response to Q2: we used a single pre-trained checkpoint and repeated the finetuning procedure three times.
> > > >
> > > > I'm not entirely clear on the why author perform multiple runs during the finetuning phase, especially when the proposed methods focus primarily on pretraining approach. To genuinely validate the stability and effectiveness of the proposed methods, wouldn't it be more appropriate to conduct multiple runs during pretraining and then finetune the resulting checkpoints? From this angle, significance of proposed methods upon baselines is still my concerns.
> > > >
> > > > > In response to Q3: We empirically found that IR slightly improves performance compared to AIR. Therefore, we choose to use SIR and our proposed AIR together.
> > > >
> > > > While the combination of improved clean performance and enhanced robustness with AIR already achieve better performance compare to IR, the empirical results stemming from the combination of IR and AIR don't seem to present a coherent strategy.
> > > >
> > > > Thus, as I remain unconvinced about the significance of the approach compared to the baselines, and I seek further clarification on the justification for the design of the regularizers, I have decided to maintain my score of 4.

---

> > > > > ### Author Response · Authors · 2023-08-19
> > > > >
> > > > > Thanks for your replies.
> > > > >
> > > > > > 1. Regarding multiple runs.
> > > > >
> > > > > First, we performed multiple runs of fine-tuning because we wanted to exempt the effect of the randomness of fine-tuning in the performance of downstream tasks. In this way, we can provide a genuine comparison of the performance of the pre-trained models.
> > > > >
> > > > > Second, to the best of our knowledge, no existing ACL work (such as ACL, AdvCL, InfoNCE, DynACL) has performed multiple runs of pre-training and then reported the error bars due to that robust pre-training is extremely time-consuming.
> > > > >
> > > > > Third, in the past two days, we conducted another two runs of ACL with IR and DynACL with IR on CIFAR-10 with different random seeds, respectively. Then, for each pre-trained checkpoint, we conducted SLF three times. Therefore, for our proposed method and each dataset, we have **nine** results of robust test accuracy and **nine** results of standard test accuracy. By conducting the t-test between the three results of the baseline (i.e., ACL and DynACL) and the **nine** results of our proposed method (shown in the below table), we find that the p-value is even smaller, which further validates the significance of our proposed method.
> > > > >
> > > > > | p-value  | AA on CIFAR-10 | SA on CIFAR-10 |
> > > > > |---|---|---|
> > > > > | ACL vs ACL with IR | 8.140e-08 | 2.723e-13 |
> > > > > | DynACL vs DynACL with IR | 1.123e-05 | 1.857e-13 |
> > > > >
> > > > > > 2. Further clarification for the design of the regularizers.
> > > > >
> > > > > SIR of paper [1] is used to regulate standard representations to be style-independent. Our proposed AIR is proposed to regulate robust representations to be style-independent. It is an intuitive way for us to combine SIR and AIR together (i.e., IR) in order to regulate both standard and robust representations.
> > > > >
> > > > > Note that ACL also uses the contrasts between standard representations (standard contrastive loss) and the contrasts between robust representations (adversarial contrastive loss) simultaneously. In a similar way, we tried to use SIR and our proposed AIR to regulate both standard and robust representations, which is a natural way for us to further improve the performance.
> > > > >
> > > > > *References*\
> > > > > [1] Representation learning via invariant causal mechanisms. Mitrovic et al., ICLR 2021.

---

### Author Rebuttal · Authors · 2023-08-04

[**Rebuttal Highlights**]

Many thanks for supportive comments from Reviewers **ZaSM** and **Ai1a**  as well as constructive comments from Reviewers **fVht** and **paeS**!

Here, we would like to demonstrate extra empirical results to resolve your concerns.

> 1. [For Reviewers **fVht** and **paeS**] In ***Table A***, we used a Student's t-test test to show that our proposed method gains **significant** improvement across tasks and datasets.

***Table A**: We report the p-value obtained by conducting a t-test between the baseline and our proposed IR corresponding to the results reported in Table 1 of the main paper.*
| p-value  | AA on CIFAR-10 | SA on CIFAR-10 | AA on CIFAR-100 | SA on CIFAR-100 | AA on STL10 | SA on STL10 |
|---|---|---|---|---|---|---|
| ACL vs ACL with IR | 9.76e-05 | 9.75e-05 | 0.0196 | 4.46e-07 | 0.0049 | 2.22e-06 |
| DynACL vs DynACL with IR | 0.0019 | 4.48e-05 | 2.22e-05 | 8.16e-06 | 0.0009 | 4.69e-09 |

According to ***Table A***, we can find that the p-value is consistently much smaller than 0.05. Therefore, our proposed method achieves significant improvement over baseline methods.

> 2. [For Reviewers **fVht** and **paeS**] In ***Table B.1*** and ***Table B.2***, we run additional experiments and provide the results of conducting DynACL with IR on **Imagenette datasets** which is a subset of ImageNet-1K.

The results validate that our proposed method is still effective.

***Table B.1**: The performance evaluated on Imagenette of ResNet-18.*
|  ResNet-18 | AA (\%) | SA (\%) |
|---|---|---|
| DynACL | 57.15 | 79.41 |
| DynACL with IR | 58.34 | 80.61 |

***Table B.2**: The performance evaluated on Imagenette of ResNet-50.*
|  ResNet-50 | AA (\%) | SA (\%) |
|---|---|---|
| DynACL | 58.98 | 80.74 |
| DynACL with IR | 60.10 | 81.66 |

To the best of our knowledge, there exits no reported result of ACL and DynACL on large-scale datasets in the existing papers. Therefore, we pre-trained ResNet-18 and ResNet-50 on Imagenette (https://github.com/fastai/imagenette) of $256 \times 256$ resolution which is a subset of ImageNet-1K using DynACL and DynACL with IR (ours), respectively. We compare and report the performance evaluated on Imagenette via SLF in ***Table B.1*** and ***Table B.2***.

Note that, due to the limitation of our GPU memory, we set the batch size to 256 and 128 for pre-training ResNet-18 and ResNet-50, respectively. We believe using a larger batch size during pre-training can even further improve the performance [1].

> 3. [For Reviewer **fVht**] In ***Table C***, we provide a comparison between DynACL with IR and supervised pre-training (SPT) including Madry and TRADES in terms of performance, pre-training time, and transferability.

***Table C**: The comparison between Madry, TRADES, and DynACL with IR.*
|  |  | SLF on CIFAR-10 | SLF on CIFAR-10 | SLF on CIFAR-100 | SLF on CIFAR-100|
|---|---|---|---|---|---|
| Pre-training on CIFAR-10 | Pre-training time (hours)| AA (\%) | SA (\%) | AA (\%) | SA (\%) |
| Madry (need labels) | 3.6 | 46.20 | 83.79 | 4.45 | 23.47 |
| TRADES  (need labels) | 5.5 | 47.50 | 84.68 | 5.23 | 22.19 |
| DynACL with IR  (no need labels)| 43.1 | 45.27 | 78.08 | 12.20 | 31.33 |

We pre-trained ResNet-18 on CIFAR-10 via Madry and TRADES. In ***Table C***, we report the standard accuracy (SA) and robust accuracy evaluated by AutoAttack (AA) on CIFAR-10 and CIFAR-100 after SLF.

(1) *Speed*. ACL-style pre-training methods do not acquire labels and thus need 1000 pre-training epochs (more time) to converge. Whereas, SPT using labels only needs 100 pre-training epochs to converge. Therefore, DynACL with IR consumes much more pre-training time compared with SPT.

(2) *Performance*. DynACL with IR achieves worse performance on CIFAR-10 than SPT via SLF. The reason could be that SPT can leverage the labels during training, thus making the feature extractor better overfit the training distribution of CIFAR-10 during pre-training. Therefore, SPT can perform better than DynACL with IR via SLF where the feature extractor is frozen.

(3) *Transferability*. DynACL with IR achieves better transferability to CIFAR-100 than SPT via SLF. Compared to DynACL with IR, SPT can use the labels and thus easily overfit the training distribution of the CIFAR-10 dataset, consequently performing worse in the generalization to other datasets such as the CIFAR-100 dataset.

> 4. [For Reviewers **fhVt** and **Ai1a**] Our proposed AIR is less sensitive to the hyper-parameter (HP) than SIR.

We provide an ablation study of $\lambda_1$ and $\lambda_2$ in Table 7. We can observe that when $\lambda_1 = 0.5$ and $\lambda_2 > 0$ (only tunning the HP of AIR), the standard deviation (STD) of the natural and robust test accuracy is 0.08 and 0.13, respectively. When $\lambda_2 = 0.5$ and $\lambda_1 > 0$ (only tunning the HP of SIR), the STD of the natural and robust test accuracy is 0.24 and 0.27, respectively. Therefore, our proposed AIR is less sensitive to the HP than SIR since AIR yields a much smaller STD of results under different HPs than SIR.

*References*

[1] Chen, Ting, et al. "A simple framework for contrastive learning of visual representations." International conference on machine learning. PMLR, 2020.

---

### Author Response · Authors · 2023-08-11
**We would like to know if you have any further questions or require additional clarification.**

Dear Reviewers,

Thank you for taking the time to review our work and for providing us with valuable feedback. We have carefully considered your comments and provided our responses.

If you have any further questions or require additional clarification, please kindly let us know.

In particular, we would like to ask Reviewers **fVht** and **paeS** if our responses satisfactorily address their concerns.

Thank you again for your valuable input.

Best wishes, \
Authors

---

### Decision · Program_Chairs · 2023-09-21

**Decision:**

Accept (poster)

**Comment:**

The reviewers acknowledge the technical novelty of the paper, which introduces an advancement in adversarial contrastive learning using causal reasoning. The proposed adversarial invariant regularization (AIR) method is commended for demonstrating a style factor and its empirical effectiveness on CIFAR10 and CIFAR100 datasets. The reviewer appreciates the authors' thorough explanation of their decision to perform multiple runs during the finetuning phase, clarifying that it aims to eliminate the impact of randomness on downstream task performance and validate the robustness of their approach. The design of the regularizers is clarified by explaining the integration of style-independent regularization (SIR) and AIR to regulate both standard and robust representations, and comparisons with ACL and DynACL are outlined. Theoretical justifications, consistent performance gains through experiments, and ablation studies on the proposed AIR term's effectiveness contribute to the paper's strength.